# FEDERATED LEARNING VIA POSTERIOR AVERAGING: A NEW PERSPECTIVE AND PRACTICAL ALGORITHMS

**Maruan Al-Shedivat**[*]  **Jennifer Gillenwater**  **Eric Xing**  **Afshin Rostamizadeh**
CMU                      Google                   MBZUAI & CMU    Google

## ABSTRACT

Federated learning is typically approached as an optimization problem, where the goal is to minimize a global loss function by distributing computation across client devices that possess local data and specify different parts of the global objective. We present an alternative perspective and formulate federated learning as a posterior inference problem, where the goal is to infer a global posterior distribution by having client devices each infer the posterior of their local data. While exact inference is often intractable, this perspective provides a principled way to search for global optima in federated settings. Further, starting with the analysis of federated quadratic objectives, we develop a computation- and communication-efficient approximate posterior inference algorithm—*federated posterior averaging* (FEDPA). Our algorithm uses MCMC for approximate inference of local posteriors on the clients and efficiently communicates their statistics to the server, where the latter uses them to refine a global estimate of the posterior mode. Finally, we show that FEDPA generalizes federated averaging (FEDAVG), can similarly benefit from adaptive optimizers, and yields state-of-the-art results on four realistic and challenging benchmarks, converging faster, to better optima.

## 1 INTRODUCTION

Federated learning (FL) is a framework for learning statistical models from heterogeneous data scattered across multiple entities (or clients) under the coordination of a central server that has no direct access to the local data (Kairouz et al., 2019). To learn models without any data transfer, clients must process their own data locally and only infrequently communicate some model updates to the server which aggregates these updates into a global model (McMahan et al., 2017). While this paradigm enables efficient distributed learning from data stored on millions of remote devices (Hard et al., 2018), it comes with many challenges (Li et al., 2020), with the communication cost often being the critical bottleneck and the heterogeneity of client data affecting convergence.

Canonically, FL is formulated as a distributed optimization problem with a few distinctive properties such as unbalanced and non-i.i.d. data distribution across the clients and limited communication. The *de facto* standard algorithm for solving federated optimization is federated averaging (FEDAVG, McMahan et al., 2017), which proceeds in rounds of communication between the server and a random subset of clients, synchronously updating the server model after each round (Bonawitz et al., 2019). By allowing the clients perform *multiple* local SGD steps (or epochs) at each round, FEDAVG can reduce the required communication by orders of magnitude compared to mini-batch (MB) SGD.

However, due to heterogeneity of the client data, more local computation often leads to biased client updates and makes FEDAVG stagnate at inferior optima. As a result, while slow during initial training, MB-SGD ends up dominating FEDAVG at convergence (see example in Fig. 1). This has been observed in multiple empirical studies (*e.g.*, Charles & Konečný, 2020), and recently was shown theoretically (Woodworth et al., 2020a). Using stateful clients (Karimireddy et al., 2019; Pathak & Wainwright, 2020) can help to remedy the convergence issues in the *cross-silo* setting, where relatively few clients are queried repeatedly, but is not practical in the *cross-device* setting (*i.e.*, when clients are mobile devices) for several reasons (Kairouz et al., 2019; Li et al., 2020; Lim et al., 2020). One key issue is that the number of clients in such a setting is extremely large and the average client will only ever participate in a single FL round. Thus, the state of a stateful algorithm is never used.

---

[*]Most of the work done at Google. Correspondence: maruan.alshedivat.com

**Figure 1:** An illustration of federated learning in a toy 2D setting with two clients and quadratic objectives. **Left:** Contour plots of the client objectives, their local optima, as well as the corresponding global optimum. **Middle:** Learning curves for MB-SGD and FEDAVG with 10 and 100 steps per round. FEDAVG makes fast progress initially, but converges to a point far away from the global optimum. **Right:** Learning curves for FEDPA with 10 and 100 posterior samples per round and shrinkage $\rho = 1$. More posterior samples (*i.e.*, more local computation) results in faster convergence and allows FEDPA to come closer to the global optimum. Shaded regions denote bootstrapped 95% CI based on 5 runs with different initializations and random seeds. Best viewed in color.

*Is it possible to design FL algorithms that exhibit both fast training and consistent convergence with stateless clients?* In this work, we answer this question affirmatively, by approaching federated learning not as optimization but rather as posterior inference problem. We show that modes of the global posterior over the model parameters correspond to the desired optima of the federated optimization objective and can be inferred by aggregating information about local posteriors. Starting with an analysis of federated quadratics, we introduce a general class of *federated posterior inference* algorithms that run local posterior inference on the clients and global posterior inference on the server. In contrast with federated optimization, posterior inference can, with stateless clients, benefit from an increased amount of local computation without stagnating at inferior optima (illustrated in Fig. 1). However, a naïve approach to federated posterior inference is practically infeasible because its computation and communication costs are cubic and quadratic in the model parameters, respectively. Apart from the new perspective, our key technical contribution is the design of an efficient algorithm with linear computation and communication costs.

**Contributions.** The main contributions of this paper can be summarized as follows:

1. We introduce a new perspective on federated learning through the lens of posterior inference which broadens the design space for FL algorithms beyond purely optimization techniques.
2. With this perspective, we design a computation- and communication-efficient approximate posterior inference algorithm—*federated posterior averaging* (FEDPA). FEDPA works with stateless clients and its computational complexity and memory footprint are similar to FEDAVG.
3. We show that FEDAVG with many local steps is in fact a special case of FEDPA that estimates local posterior covariances with identities. These biased estimates are the source of inconsistent updates and explain why FEDAVG has suboptimal convergence even in simple quadratic settings.
4. Finally, we compare FEDPA with strong baselines on realistic FL benchmarks introduced by Reddi et al. (2020) and achieve state-of-the-art results with respect to multiple metrics of interest.

## 2 RELATED WORK

**Federated optimization.** Starting with the seminal paper by McMahan et al. (2017), a lot of recent effort in federated learning has focused on understanding of FEDAVG (also known as local SGD) as an optimization algorithm. Multiple works have provided upper bounds on the convergence rate of FEDAVG in the homogeneous i.i.d. setting (Yu et al., 2019; Karimireddy et al., 2019; Woodworth et al., 2020b) as well as explored various non-i.i.d. settings with different notions of heterogeneity (Zhao et al., 2018; Sahu et al., 2018; Hsieh et al., 2019; Li et al., 2019; Wang et al., 2020; Woodworth et al., 2020a). Reddi et al. (2020) reformulated FEDAVG in a way that enabled adaptive optimization and derived corresponding convergence rates, noting that FEDAVG requires careful tuning of learning rate schedules in order to converge to the desired optimum, which was further analyzed by Charles & Konečný (2020). To the best of our knowledge, our work is perhaps the first to connect, reinterpret, and analyze federated optimization from the probabilistic inference perspective.

**Distributed MCMC.** Part of our work builds on the idea of sub-posterior aggregation, which was originally proposed for scaling up Markov chain Monte Carlo techniques to large datasets (known as the *concensus Monte Carlo*, Neiswanger et al., 2013; Scott et al., 2016). One of the goals of this paper is to highlight the connection between distributed inference and federated optimization and develop inference techniques that can be used under FL-specific constraints.

## 3 A POSTERIOR INFERENCE PERSPECTIVE ON FEDERATED LEARNING

Federated learning is typically formulated as the following optimization problem:

$$\min_{\boldsymbol{\theta} \in \mathbb{R}^d} \left\{ F(\boldsymbol{\theta}) := \sum_{i=1}^{N} q_i f_i(\boldsymbol{\theta}) \right\}, \quad f_i(\boldsymbol{\theta}) := \frac{1}{n_i} \sum_{j=1}^{n_i} f(\boldsymbol{\theta}; z_{ij}), \tag{1}$$

where the global objective function $F(\boldsymbol{\theta})$ is a weighted average of the local objectives $f_i(\boldsymbol{\theta})$ over $N$ clients; each client's objective is some loss $f(\boldsymbol{\theta}; z)$ computed on the local data $D_i = \{z_{i1}, \ldots, z_{in_i}\}$. In real-world cross-device applications, the total number of clients $N$ can be extremely large, and hence optimization of $F(\boldsymbol{\theta})$ is done over multiple rounds with only a small subset of $M$ clients participating in each round. The weights $\{q_i\}$ are typically set proportional to the sizes of the local datasets $\{n_i\}$, which makes $F(\boldsymbol{\theta})$ coincide with the training objective of the centralized setting.

Typically, $f(\boldsymbol{\theta}; z)$ is negative log likelihood of $z$ under some probabilistic model parametrized by $\boldsymbol{\theta}$, *i.e.*, $f(\boldsymbol{\theta}; z) := -\log \mathbb{P}(z \mid \boldsymbol{\theta})$. For example, least squares loss corresponds to likelihood under a Gaussian model, cross entropy loss corresponds to likelihood under a categorical model, etc. (Murphy, 2012). Thus, Eq. 1 corresponds to *maximum likelihood estimation* (MLE) of the model parameters $\boldsymbol{\theta}$.

An alternative (Bayesian) approach to maximum likelihood estimation is *posterior inference* or estimation of the posterior distribution of the parameters given all the data: $\mathbb{P}(\boldsymbol{\theta} \mid D \equiv D_1 \cup \cdots \cup D_N)$. The posterior is proportional to the product of the likelihood and a prior, $\mathbb{P}(\boldsymbol{\theta} \mid D) \propto \mathbb{P}(D \mid \boldsymbol{\theta}) \mathbb{P}(\boldsymbol{\theta})$, and, if the prior is uninformative (uniform over all $\boldsymbol{\theta}$), the modes of the global posterior coincide with MLE solutions or optima of $F(\boldsymbol{\theta})$ in Eq. 1. While this simple observation establishes an equivalence between the inference of the posterior mode and optimization, the advantage of this perspective comes from the fact that the global posterior *exactly* decomposes into a product of local posteriors.[1]

**Proposition 1 (Global Posterior Decomposition)** *Under the uniform prior, any global posterior distribution that exists decomposes into a product of local posteriors:* $\mathbb{P}(\boldsymbol{\theta} \mid D) \propto \prod_{i=1}^{N} \mathbb{P}(\boldsymbol{\theta} \mid D_i)$.

Proposition 1 suggests that as long as we are able to compute local posterior distributions $\mathbb{P}(\boldsymbol{\theta} \mid D_i)$ and communicate them to the server, we should be able to solve Eq. 1 by multiplicatively aggregating them to find the mode of the global posterior $\mathbb{P}(\boldsymbol{\theta} \mid D)$ on the server. Note that posterior inference via multiplicative averaging has been successfully used to scale Monte Carlo methods to large datasets, where the approach is *embarrassingly parallel* (Neiswanger et al., 2013; Scott et al., 2016). In the FL context, this means that once all clients have sent their local posteriors to the server, we can construct the global posterior without any additional communication. However, there remains the challenge of making the local and global inference and communication efficient enough for real federated settings. The example below illustrates how this can be difficult even for a simple model and loss function.

**Federated least squares.** Consider federated least squares regression with a linear model, where $z := (\mathbf{x}, y)$ and the loss $f(\boldsymbol{\theta}; \mathbf{x}, y) := \frac{1}{2}(\mathbf{x}^\top \boldsymbol{\theta} - y)^2$ is quadratic. Then, the client objective becomes:

$$f_i(\boldsymbol{\theta}) = \log \exp \left\{ \frac{1}{2} \|\mathbf{X}_i \boldsymbol{\theta} - \mathbf{y}_i\|^2 \right\} = \log \exp \left\{ \frac{1}{2}(\boldsymbol{\theta} - \boldsymbol{\mu}_i)^\top \boldsymbol{\Sigma}_i^{-1} (\boldsymbol{\theta} - \boldsymbol{\mu}_i) \right\} + \text{const}, \tag{2}$$

where $\mathbf{X}_i \in \mathbb{R}^{n_i \times d}$ is the design matrix, $\mathbf{y}_i \in \mathbb{R}^{n_i}$ is the response vector, $\boldsymbol{\Sigma}_i^{-1} := \mathbf{X}_i^\top \mathbf{X}_i$ and $\boldsymbol{\mu}_i := (\mathbf{X}_i^\top \mathbf{X}_i)^{-1} \mathbf{X}_i^\top \mathbf{y}_i$. Note that the expression in Eq. 2 is the log likelihood for a multivariate Gaussian distribution with mean $\boldsymbol{\mu}_i$ and covariance $\boldsymbol{\Sigma}_i$. Therefore, each local posterior (under the uniform prior) is Gaussian, and, as a product of Gaussians, the global posterior is also Gaussian with the following mean (which coincides with the posterior mode):

$$\boldsymbol{\mu} := \left( \sum_{i=1}^{N} q_i \boldsymbol{\Sigma}_i^{-1} \right)^{-1} \left( \sum_{i=1}^{N} q_i \boldsymbol{\Sigma}_i^{-1} \boldsymbol{\mu}_i \right). \tag{3}$$

Concretely, in the case of least squares regression, this suggests that it is sufficient for clients to infer the means $\{\boldsymbol{\mu}_i\}$ and inverse covariances $\{\boldsymbol{\Sigma}_i^{-1}\}$ of their local posteriors and communicate that information to server for the latter to be able to find the global optimum. However, a straightforward application of Eq. 3 would require $\mathcal{O}(d^2)$ space and $\mathcal{O}(d^3)$ computation, both on the clients and on the server, which is very expensive for the typical cross-device FL setting. Similarly, the communication cost would be $\mathcal{O}(d^2)$, while standard FL algorithms have communication cost of $\mathcal{O}(d)$.

---

[1]Note that from the optimization point of view, the global optimum generally cannot be represented as any weighted combination of the local optima even in simple 2D settings (see Fig. 1, left).

| **Algorithm 1** Generalized Federated Optimization | **Algorithm 2** Client Update (FEDAVG) |
|---|---|

**Algorithm 1** Generalized Federated Optimization

**input** initial $\boldsymbol{\theta}$, CLIENTUPDATE, SERVERUPDATE
1: **for** each round $t = 1, \ldots, T$ **do**
2:     Sample a subset $\mathcal{S}$ of clients
3:     **communicate** $\boldsymbol{\theta}$ to all $i \in \mathcal{S}$ // server → clients
4:     **for** each client $i \in \mathcal{S}$ **in parallel do**
5:         $\boldsymbol{\Delta}_i^t, q_i \leftarrow$ CLIENTUPDATE$(\boldsymbol{\theta})$
6:     **end for**
7:     **communicate** $\{\boldsymbol{\Delta}_i^t, q_i\}_{i \in \mathcal{S}}$   // server ← clients
8:     $\boldsymbol{\Delta}^t \leftarrow \frac{1}{|\mathcal{S}|} \sum_{i \in \mathcal{S}} q_i \boldsymbol{\Delta}_i^t$   // aggregate updates
9:     $\boldsymbol{\theta} \leftarrow$ SERVERUPDATE$(\boldsymbol{\theta}, \boldsymbol{\Delta}^t)$
10: **end for**
**output** final $\boldsymbol{\theta}$

**Algorithm 2** Client Update (FEDAVG)

**input** initial $\boldsymbol{\theta}_0$, loss $f_i(\boldsymbol{\theta})$, optimizer CLIENTOPT
1: **for** $k = 1, \ldots, K$ **do**
2:     $\boldsymbol{\theta}_k \leftarrow$ CLIENTOPT$(\boldsymbol{\theta}_{k-1}, \hat{\nabla} f_i(\boldsymbol{\theta}_{k-1}))$
3: **end for**
**output** $\boldsymbol{\Delta} := \boldsymbol{\theta}_0 - \boldsymbol{\theta}_k$, client weight $q_i$

**Algorithm 3** Client Update (FEDPA)

**input** initial $\boldsymbol{\theta}_0$, loss $f_i(\boldsymbol{\theta})$, sampler CLIENTMCMC
1: **for** $k = 1, \ldots, K$ **do**
2:     $\boldsymbol{\theta}_k \sim$ CLIENTMCMC$(\boldsymbol{\theta}_{k-1}, f_i)$
3: **end for**
**output** $\boldsymbol{\Delta} := \hat{\boldsymbol{\Sigma}}^{-1}(\boldsymbol{\theta}_0 - \hat{\boldsymbol{\mu}})$, client weight $q_i$

**Approximate federated posterior inference.** Apart from the computation and communication issues discussed in the simple example above, we also have to contend with the fact that, generally, posteriors are non-Gaussian and closed form expressions for global posterior modes may not exist.[2] In such cases, we propose to use the *Laplace approximation for local and global posteriors*, *i.e.*, approximate them with the best-fitting Gaussians. While imperfect, this approximation will allow us to compute the (approximate) global posterior mode in a computation- and communication-efficient manner using the following three steps: (i) infer approximate local means $\{\hat{\boldsymbol{\mu}}_i\}$ and covariances $\{\hat{\boldsymbol{\Sigma}}_i\}$, (ii) communicate these to the server, and (iii) compute the posterior mode given by Eq. 3. Note that directly computing and communicating these quantities would be completely infeasible for the realistic setting where models are neural networks with millions of parameters. In the following section, we design a practical algorithm where all costs are linear in the number of model parameters.

## 4 FEDERATED POSTERIOR AVERAGING: A PRACTICAL ALGORITHM

Federated averaging (FEDAVG, McMahan et al., 2017) solves the problem from Eq. 1 over $T$ rounds by interacting with $M$ random clients at each round in the following way: (i) broadcasting the current model parameters $\boldsymbol{\theta}$ to the clients, (ii) running SGD for $K$ steps on each client, and (iii) updating the global model parameters by collecting and averaging the final SGD iterates. Reddi et al. (2020) reformulated the same algorithm in the form of server- and client-level optimization (Algorithm 1), which allowed them to bring techniques from the adaptive optimization literature to FL.

FEDAVG is efficient in that it requires only $\mathcal{O}(d)$ computation on both the clients and the server, and $\mathcal{O}(d)$ communication between each client and the server. To arrive at a similarly efficient algorithm for posterior inference, we focus on the following questions: (a) how to estimate local and global posterior moments efficiently? (b) how to communicate local statistics to the server efficiently?

**(1) Efficient global posterior inference.** There are two issues with computing an estimate of the global posterior mode $\boldsymbol{\mu}$ directly using Eq. 3. First, it requires computing the inverse of a $d \times d$ matrix on the server, which is an $\mathcal{O}(d^3)$ operation. Second, it relies on acquiring local means and inverse covariances, which would require $\mathcal{O}(d^2)$ communication from each client. We propose to solve both issues by converting the global posterior estimation into an equivalent optimization problem.

**Proposition 2 (Global Posterior Inference)** *The global posterior mode $\boldsymbol{\mu}$ given in Eq. 3 is the minimizer of a quadratic $\mathcal{Q}(\boldsymbol{\theta}) := \frac{1}{2}\boldsymbol{\theta}^\top \mathbf{A}\boldsymbol{\theta} - \mathbf{b}^\top \boldsymbol{\theta}$, where $\mathbf{A} := \sum_{i=1}^N q_i \boldsymbol{\Sigma}_i^{-1}$ and $\mathbf{b} := \sum_{i=1}^N q_i \boldsymbol{\Sigma}_i^{-1} \boldsymbol{\mu}_i$.*

Proposition 2 allows us to obtain a good estimate of $\boldsymbol{\mu}$ by running stochastic optimization of the quadratic objective $\mathcal{Q}(\boldsymbol{\theta})$ on the server. Note that the gradient of $\mathcal{Q}(\boldsymbol{\theta})$ has the following form:

$$\nabla \mathcal{Q}(\boldsymbol{\theta}) := \sum_{i=1}^N q_i \boldsymbol{\Sigma}_i^{-1}(\boldsymbol{\theta} - \boldsymbol{\mu}_i), \tag{4}$$

which suggests that we can obtain $\boldsymbol{\mu}$ by using the same Algorithm 1 as FEDAVG but using different client updates: $\boldsymbol{\Delta}_i := \boldsymbol{\Sigma}_i^{-1}(\boldsymbol{\theta} - \boldsymbol{\mu}_i)$. Importantly, as long as clients are able to compute $\boldsymbol{\Delta}_i$'s, this approach will result in $\mathcal{O}(d)$ communication and $\mathcal{O}(d)$ server computation cost per round.

---

[2]Gaussian posteriors can be further generalized to the exponential family for which closed form expressions can be obtained under appropriate priors (Wainwright & Jordan, 2008). We leave this extension to future work.

**(2) Efficient local posterior inference.** To compute $\mathbf{\Delta}_i$, each client needs to be able to estimate the local posterior means and covariances. We propose to use stochastic gradient Markov chain Monte Carlo (SG-MCMC, Welling & Teh, 2011; Ma et al., 2015) for approximate sampling from local posteriors on the clients, so that these samples can be used to estimate $\hat{\boldsymbol{\mu}}_i$'s and $\hat{\mathbf{\Sigma}}_i$'s. Specifically, we use a variant of SG-MCMC[3] with iterate averaging (IASG, Mandt et al., 2017), which involves: (a) running local SGD for some number of steps to mix in the Markov chain, then (b) continued running of SGD for more steps to periodically produce samples via Polyak averaging (Polyak & Juditsky, 1992) of the intermediate iterates (Algorithm 4). The more computation we can run locally on the clients each round, the more posterior samples can be produced, resulting in better estimates of the local moments.

---

**Algorithm 4** IASG Sampling (CLIENTMCMC)

**input** initial $\boldsymbol{\theta}$, loss $f_i(\boldsymbol{\theta})$, optimizer CLIENTOPT($\alpha$), $B$: burn-in steps, $K$: steps per sample, $\ell$: # samples.

    // Burn-in
1: **for** step $t = 1, \ldots, B$ **do**
2:     $\boldsymbol{\theta} \leftarrow$ CLIENTOPT($\boldsymbol{\theta}, \hat{\nabla} f_i(\boldsymbol{\theta})$)
3: **end for**
    // Sampling
4: **for** sample $s = 1, \ldots, \ell$ **do**
5:     $\mathcal{S}_{\boldsymbol{\theta}} \leftarrow \varnothing$         // Initialize iterates
6:     **for** step $t = 1, \ldots, K$ **do**
7:         $\boldsymbol{\theta} \leftarrow$ CLIENTOPT($\boldsymbol{\theta}, \hat{\nabla} f_i(\boldsymbol{\theta})$)
8:         $\mathcal{S}_{\boldsymbol{\theta}} \leftarrow \mathcal{S}_{\boldsymbol{\theta}} \cup \{\boldsymbol{\theta}\}$
9:     **end for**
10:    $\boldsymbol{\theta}_s \leftarrow$ AVERAGE($\mathcal{S}_{\boldsymbol{\theta}}$)     // Average iterates
11: **end for**
**output** samples $\{\boldsymbol{\theta}_1, \ldots, \boldsymbol{\theta}_\ell\}$

---

**(3) Efficient computation of the deltas.** Even if we can obtain samples $\{\hat{\boldsymbol{\theta}}_1, \ldots, \hat{\boldsymbol{\theta}}_\ell\}$ via MCMC and use them to estimate local moments, $\hat{\boldsymbol{\mu}}_i$ and $\hat{\mathbf{\Sigma}}_i$, computing $\mathbf{\Delta}_i$ naïvely would still require inverting a $d \times d$ matrix, *i.e.*, $\mathcal{O}(d^3)$ compute and $\mathcal{O}(d^2)$ memory. The good news is that we are able to show that clients can compute $\mathbf{\Delta}_i$'s much more efficiently, in $\mathcal{O}(d)$ time and memory, using a dynamic programming algorithm and appropriate mean and covariance estimators.

**Theorem 3** *Given $\ell$ approximate posterior samples $\{\hat{\boldsymbol{\theta}}_1, \ldots, \hat{\boldsymbol{\theta}}_\ell\}$, let $\hat{\boldsymbol{\mu}}_\ell$ be the sample mean, $\hat{\mathbf{S}}_\ell$ be the sample covariance, and $\hat{\mathbf{\Sigma}}_\ell := \rho_\ell \mathbf{I} + (1 - \rho_\ell)\hat{\mathbf{S}}_\ell$ be a shrinkage estimator (Ledoit & Wolf, 2004b) of the covariance with $\rho_\ell := 1/(1 + (\ell - 1)\rho)$ for some $\rho \in [0, +\infty)$. Then, for any $\boldsymbol{\theta}$, we can compute $\hat{\mathbf{\Delta}}_\ell = \hat{\mathbf{\Sigma}}_\ell^{-1}(\boldsymbol{\theta} - \hat{\boldsymbol{\mu}}_\ell)$ in $\mathcal{O}(\ell^2 d)$ time and using $\mathcal{O}(\ell d)$ memory.*

**Proof** [Sketch] We give a constructive proof by designing an efficient algorithm for computing $\hat{\mathbf{\Delta}}_\ell$. Our approach is based on two key ideas:

1. We prove that the specified shrinkage estimator of the covariance has a recursive decomposition into rank-1 updates, *i.e.*, $\hat{\mathbf{\Sigma}}_t = \hat{\mathbf{\Sigma}}_{t-1} + c_t \cdot \mathbf{x}_t^\top \mathbf{x}_t$, where $c_t$ is a constant and $\mathbf{x}_t$ is some vector. This allows us to leverage the Sherman-Morrison formula for computing the inverse of $\hat{\mathbf{\Sigma}}_\ell$.

2. Further, we design a dynamic programming algorithm for computing $\hat{\mathbf{\Delta}}_\ell$ exactly without storing the covariance matrix or its inverse. Our algorithm is online and allows efficient updates of $\hat{\mathbf{\Delta}}_\ell$ as more posterior samples become available.

See Appendix C for the full proof and derivation of the algorithm. ∎

Note that the computational cost of $\hat{\mathbf{\Delta}}_\ell$ consists of two components: (i) the cost of producing $\ell$ approximate local posterior samples using IASG and (ii) the cost of solving a linear system using dynamic programming. How much of an overhead does it add compared to simply running local SGD? It turns out that in practical settings the overhead is almost negligible. Table 1 shows the time it takes a client to compute the updates based on 5 local epochs (100 steps per epoch) using different

**Table 1:** Computational complexity of the client updates for methods that use 5 local epochs measured in milliseconds (% denotes relative increase).

| Dim | $\hat{\mathbf{\Delta}}_{\text{FEDAVG}}$ | $\hat{\mathbf{\Delta}}_\ell$ (DP) | | $\hat{\mathbf{\Delta}}_\ell$ (exact) | |
|---|---|---|---|---|---|
| 100 | 72 | 91 | +26% | 82 | +12% |
| 1K | 76 | 92 | +21% | 104 | +36% |
| 10K | 80 | 93 | +16% | 797 | +896% |
| 100K | 149 | 155 | +4% | | — |

algorithms (FEDAVG vs. our approach with exact or dynamic programming (DP) matrix inversion) on synthetic linear regressions. As the dimensionality grows, computational complexity of DP-based estimation of $\hat{\mathbf{\Delta}}_\ell$ becomes nearly identical to FEDAVG, which indicates that the majority of the cost in practice would come from SGD steps rather than our dynamic programming procedure.

---

[3]While in this work we use a variant of SG-MCMC, other techniques such as HMC (Neal et al., 2011) or NUTs (Hoffman & Gelman, 2014) can be used, too. We leave analysis of alternative approaches to future work.

**The final algorithm, discussion, and implications.** Putting all the pieces together, we arrive at the *federated posterior averaging* (FEDPA) algorithm for approximately computing the mode of the global posterior over multiple communication rounds. Our algorithm is a variant of generalized federated optimization (Algorithm 1) with a new client update procedure (Algorithm 3). Importantly, this also implies that FEDAVG can be viewed as posterior inference algorithm that estimates $\hat{\boldsymbol{\Sigma}}$ with an identity and, as a result, obtains biased client deltas $\hat{\boldsymbol{\Delta}}_{\text{FEDAVG}} := \mathbf{I}(\boldsymbol{\theta} - \hat{\boldsymbol{\mu}})$.

In Fig. 1 in the introduction, we demonstrate the differences in behavior between FEDAVG and FEDPA that stem from the differences in their client updates. Biased client updates make FEDAVG converge to a suboptimal point; moreover, increasing local computation only pushes the fixed point further away from the global optimum. On the other hand, FEDPA converges faster and to a better optimum, trading off bias for slightly more variance (becomes visible only closer to convergence). We see that FEDPA also substantially benefits from more local computation (more local samples).

Since the main difference between FEDAVG and FEDPA is, in fact, the bias-variance trade off in the server gradient estimates (Eq. 4), we can view both methods as *biased* SGD (Ajalloeian & Stich, 2020) and reason about their convergence rates as well as distances between their fixed points and correct global optima as functions of the gradient bias. In Appendix A, we provide further details, discuss convergence, empirically quantify the bias and variance of the client updates for both methods, and analyse the effects of the sampling-based approximations on the behavior of FEDPA.

## 5 EXPERIMENTS

Using a suite of realistic benchmark tasks introduced by Reddi et al. (2020), we evaluate FEDPA against several competitive baselines: the best versions of FEDAVG with adaptive optimizers as well as MIME (Karimireddy et al., 2020)—a recently-proposed FEDAVG variant that also works with stateless clients, but uses control-variates and server-level statistics to mitigate convergence issues.

**Table 2:** Statistics on the data and tasks. The number of examples per client are given with one standard deviation across the corresponding set of clients (denoted with $\pm$). See description of the tasks in the text.

| Dataset | Task | # classes | # clients (train/test) | # examples p/ client (train/test) |
|---|---|---|---|---|
| EMNIST-62 | CR | 62 | 3,400 / 3,400 | $198 \pm 77$ / $23 \pm 9$ |
| CIFAR-100 | IR | 100 | 500 / 100 | $100 \pm 0$ / $100 \pm 0$ |
| StackOverflow | LR
NWP | 500
10,000 | 342,477 / 204,088 | $397 \pm 1279$ / $81 \pm 301$ |

### 5.1 THE SETUP

**Datasets and tasks.** The four benchmark tasks are based on the following three datasets (Table 2): EMNIST (Cohen et al., 2017), CIFAR100 (Krizhevsky et al., 2009), and StackOverflow (StackOverflow, 2016). EMNIST (handwritten characters) and CIFAR100 (RGB images) are used for multi-class image classification tasks. StackOverflow (text) is used for next-word prediction (also a multi-class classification task, historically denoted NWP) and tag prediction (a multi-label classification task, historically denoted LR because a logistic regression model is used). EMNIST was partitioned by authors (Caldas et al., 2018), CIFAR100 was partitioned randomly into 600 clients with a realistic heterogeneous structure (Reddi et al., 2020), and StackOverflow was partitioned by its unique users. All datasets were preprocessed using the code provided by Reddi et al. (2020).

**Methods and models.** We use a generalized framework for federated optimization (Algorithm 1), which admits arbitrary adaptive server optimizers and expects clients to compute model deltas. As a baseline, we use federated averaging with adaptive optimizers (or with momentum) on the server and refer to it as FEDAVG-1E or FEDAVG-ME, which stands for 1 or multiple local epochs performed by clients at each round, respectively.[4] The number of local epochs in the multi-epoch versions is a hyperparameter. We use the same framework for federated posterior averaging and refer to it as FEDPA-ME. As our clients use IASG to produce approximate posterior samples, collecting

---

[4]Reddi et al. (2020) referred to federated averaging with adaptive server optimizers as FEDADAM, FEDYOGI, etc. Instead, we select the best optimizer for each task and refer to the corresponding method simply as FEDAVG.

**(a)** CIFAR-100: Evaluation loss (left) and accuracy (right) for FEDAVG-ME and FEDPA-ME.

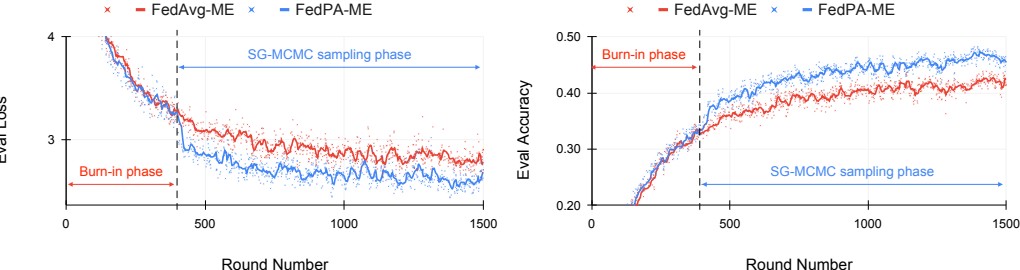

**(b)** StackOverflow LR: Evaluation loss (left) and macro-F1 (right) for FEDAVG-ME and FEDPA-ME.

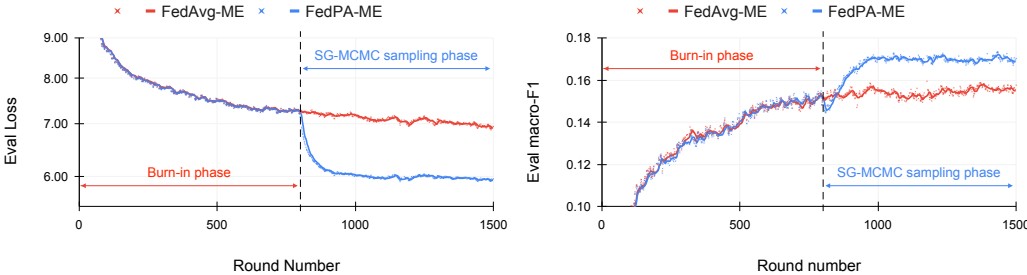

**Figure 2:** Evaluation metrics for FEDAVG and FEDPA computed at each training round on (a) CIFAR-100 and (b) StackOverflow LR. During the initial rounds (the "burn-in phase"), FEDPA computes deltas the same way as FEDAVG; after that, FEDPA computes deltas using Algorithm 3 and approximate posterior samples.

a single sample per epoch is optimal (Mandt et al., 2017). Thus FEDPA-ME uses M samples to estimate client deltas and has the same local and global computational complexity as FEDAVG-ME but with two extra hyperparameters: the number of burn-in rounds and the shrinkage coefficient $\rho$ from Theorem 3. As in Reddi et al. (2020), we use the following model architectures for each task: CNN for EMNIST-62, ResNet-18 for CIFAR-100, LSTM for StackOverflow NWP, and multi-label logistic regression on bag-of-words vectors for StackOverflow LR (for details see Appendix D).

**Hyperparameters.** For hyperparameter tuning, we first ran small grid searches for FEDAVG-ME using the best server optimizer and corresponding learning rate grids from Reddi et al. (2020). Then, we used the best FEDAVG-ME configuration and did a small grid search to tune the additional hyperparameters of FEDPA-ME, which turned out not to be very sensitive (*i.e.*, many configurations provided results superior to FEDAVG). More hyperparameter details can be found in Appendix D.

**Metrics.** Since both speed of learning as well as final performance are important quantities for federated learning, we measure: (i) the number of rounds it takes the algorithm to attain a desired level of an evaluation metric and (ii) the best performance attained within a specified number of rounds. For EMNIST-62, we measure the number of rounds it takes different methods to achieve 84% and 86% evaluation accuracy[5], and the best validation accuracy attained within 500 and 1500 rounds. For CIFAR-100, we use the same metrics but use 30% and 40% as evaluation accuracy cutoffs and 1000 and 1500 as round number cutoffs. Finally, for StackOverflow, we measure the the number of rounds it takes to the best performance and evaluation accuracy (for the NWP task) and precision, recall at 5, macro- and micro-F1 (for the LR task) attained by round 1500. We note that the total number of rounds was selected based on computational considerations (to ensure reproducibility within a reasonable amount of computational cost) and the intermediate cutoffs were selected qualitatively to highlight some performance points of interest. In addition, we provide plots of the evaluation loss and other metrics for all methods over the course of training which show a much fuller picture of the behavior of the algorithms (most of the plots are given in Appendix E).

**Implementation and reproducibility.** All our experiments on the benchmark tasks were conducted in simulation using TensorFlow Federated (TFF, Ingerman & Ostrowski, 2019). Synthetic experiments were conducted using JAX (Bradbury et al., 2018). The JAX implementation of the algorithms is available at `https://github.com/alshedivat/fedpa`. The TFF implementation will be released through `https://github.com/google-research/federated`.

---

[5]Centralized optimization of the CNN model on EMNIST-62 attains the evaluation accuracy of 88%.

**Table 3:** Comparison of FEDPA with baselines. All metrics were computed on the evaluation sets and averaged over the last 100 rounds before the round limit was reached. The "number of rounds to accuracy" was determined based on the 10-round running average crossing the threshold for the first time. The arrows indicate whether higher (↑) or lower (↓) is better. The best performance in each column is denoted in **bold**.

**(a)** EMNIST-62

| Method \@ | accuracy (%, ↑) | | rounds (#, ↓) | |
| --- | --- | --- | --- | --- |
| | 500R | 1500R | 84% | 86% |
| AFO † | 80.4 | 86.8 | 546 | 1291 |
| MIME ‡ | 83.1 | *84.9 | 464 | *— |
| FEDAVG-1E | 83.9 | 86.5 | 451 | 1360 |
| FEDAVG-ME | 85.8 | 85.9 | 86 | — |
| FEDPA-ME | **86.5** | **87.3** | **84** | **92** |

**(b)** CIFAR-100

| Method \@ | accuracy (%, ↑) | | rounds (#, ↓) | |
| --- | --- | --- | --- | --- |
| | 1000R | 1500R | 30% | 40% |
| AFO † | 31.9 | 41.1 | 898 | 1401 |
| MIME ‡ | 33.2 | *33.9 | 680 | *— |
| FEDAVG-1E | 24.2 | 31.7 | 1379 | — |
| FEDAVG-ME | 40.2 | 42.1 | **348** | 896 |
| FEDPA-ME | **44.3** | **46.3** | 348 | **543** |

**(c)** StackOverflow

| Method \ Metric | NWP | | LR (all metrics in %, ↑) | | | |
| --- | --- | --- | --- | --- | --- | --- |
| | accuracy (%, ↑) | rounds (#, ↓) | precision | recall@5 | ma-F1 | mi-F1 |
| AFO † | **23.4** | 1049 | — | 68.0 | — | — |
| FEDAVG-1E | 22.8 | 1074 | 74.58 | **69.1** | 14.9 | 43.8 |
| FEDAVG-ME | 23.0 | 870 | **78.65** | 68.7 | 15.6 | 43.3 |
| FEDPA-ME | **23.4** | **805** | 72.8 | 68.6 | **17.3** | **44.0** |

† the best results taken from (Reddi et al., 2020). ‡ the best results taken from (Karimireddy et al., 2020).
* results were only available for the method trained to 1000 rounds.

## 5.2 RESULTS ON BENCHMARK TASKS

**The effects of posterior correction of client deltas.** As we demonstrated in Section 4, FEDPA essentially generalizes FEDAVG and only differs in the computation done on the clients, where we compute client deltas using an estimator of the local posterior inverse covariance matrix, $\Sigma_i^{-1}$, which requires sampling from the posterior. To be able to use SG-MCMC for local sampling, we first run FEDPA in the *burn-in regime* (which is identical to FEDAVG) for a number of rounds to bring the server state closer to the clients' local optima,[6] after which we "turn on" the local posterior sampling. The effect of switching from FEDAVG to FEDPA for CIFAR-100 (after 400 burn-in rounds) and StackOverflow LR (after 800 burn-in rounds) is presented on Figs. 2a and 2b, respectively.[7] During the burn-in phase, evaluation performance is identical for both methods, but once FEDPA starts computing client deltas using local posterior samples, the loss immediately drops and the convergence trajectory changes, indicating that FEDPA is able to avoid stagnation and make progress towards a better optimum. Similar effects are observed across all other tasks (see Appendix E).[8]

While the improvement of FEDPA over FEDAVG on some of the tasks is visually apparent (Fig. 2), we provide a more detailed comparison of the methods in terms of the speed of learning and the attained performance on all four benchmark tasks, summarized in Table 3 and discussed below.

**Results on EMNIST-62 and CIFAR-100.** In Tables 3a and 3b, we present a comparison of FEDPA against: tuned FEDAVG with a fixed client learning rate (denoted FEDAVG-1E and FEDAVG-ME), the best variation of adaptive FEDAVG from Reddi et al. (2020) with exponentially decaying client learning rates (denoted AFO), and MIME of Karimireddy et al. (2020). With more local epochs, we see significant improvement in terms of speed of learning: both FEDPA-ME and FEDAVG-ME achieve 84% accuracy on EMNIST-62 in under 100 rounds (similarly, both methods attain 30% on CIFAR-100 by round 350). However, more local computation eventually hurts FEDAVG leading to

---

[6]If SGD cannot reach the vicinity of clients' local optima within the specified number of local steps or epochs, estimated local means and covariances based on the SGD iterates can be arbitrarily poor.

[7]The number of burn-in rounds is a hyperparamter and was selected for each task to maximize performance. See more details in Appendix D.

[8]We note that running burn-in for a fixed number of rounds before switching to sampling was a design choice; other, more adaptive strategies for determining when to switch from burn-in to sampling are certainly possible (*e.g.*, use local loss values to determine when to start sampling). We leave such alternatives as future work.

worse optima: on EMNIST-62, FEDAVG-ME is not able to consistently achieve 86% accuracy within 1500 rounds; on CIFAR-100, it takes extra 350 rounds for FEDAVG-ME to get to 40% accuracy.

Finally, federated posterior averaging achieves the best performance on both tasks in terms of evaluation accuracy within the specified limit on the number of training rounds. On EMNIST-62 in particular, the final performance of FEDPA-ME after 1500 training rounds is 87.3%, which, while only a 0.5% absolute improvement, bridges **41.7%** of the gap between the centralized model accuracy (88%) and the best federated accuracy from previous work (86.8%, Reddi et al., 2020).

**Results on StackOverflow NWP and LR.** Results for StackOverflow are presented in Table 3c. Although not as pronounced as for image datasets, we observe some improvement of FEDPA over FEDAVG here as well. For NWP, we have an accuracy gain of 0.4% over the best baseline. For the LR task, we compare methods in terms of average precision, recall at 5, and macro-/micro-F1. The first two metrics have appeared in some prior FL work, while the latter two are the primary evaluation metrics typically used in multi-label classification work (Gibaja & Ventura, 2015). Interestingly, while FEDPA underperforms in terms of precision and recall, it substantially outperforms in terms of micro- and macro-averaged F1, especially the macro-F1. This indicates that while FEDAVG learns a model that can better predict high-frequency labels, FEDPA learns a model that better captures rare labels (Yang, 1999; Yang & Liu, 1999). Interestingly, note while FEDPA improves on F1 metrics and has almost the same recall at 5, it's precision after 1500 rounds is worse than FEDAVG. A more detailed discussion along with training curves for each evaluation metric are provided in Appendix E.

## 6 CONCLUSION AND FUTURE DIRECTIONS

In this work, we presented a new perspective on federated learning based on the idea of global posterior inference via averaging of local posteriors. Applying this perspective, we designed a new algorithm that generalizes federated averaging, is similarly practical and efficient, and yields state-of-the-art results on multiple challenging benchmarks. While our algorithm required a number of specific approximation and design choices, we believe that the underlying approach has potential to significantly broaden the design space for FL algorithms beyond purely optimization techniques.

**Limitations and future work.** As we mentioned throughout the paper, our method has a number of limitations due to the design choices, such as specific posterior sampling and covariance estimation techniques. While in the appendix we analyzed the effects of some of these design choices, exploration of: (i) other sampling strategies, (ii) more efficient covariance estimators (Hsieh et al., 2013), (iii) alternatives to MCMC (such as variational inference), and (iv) more general connections with Bayesian deep learning are all interesting directions to pursue next. Finally, while there is a known, interesting connection between posterior sampling and differential privacy (Wang et al., 2015), better understanding of privacy implications of posterior inference in federated settings is an open question.

### ACKNOWLEDGMENTS

The authors would like to thank Zachary Charles for the invaluable feedback that influenced the design of the methods and experiments, and Brendan McMahan, Zachary Garrett, Sean Augenstein, Jakub Konečný, Daniel Ramage, Sanjiv Kumar, Sashank Reddi, Jean-François Kagy for many insightful discussions, and Willie Neiswanger for helpful comments on the early drafts.

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

## A  PRELIMINARY ANALYSIS AND ABLATIONS

In Section 4, we derived federated posterior averaging (FEDPA) starting with the global posterior decomposition (Proposition 1, which is exact) and applying the following three approximations:

1. The Laplace approximation of the local and global posterior distributions.
2. The shrinkage estimation of the local moments.
3. Approximate sampling from the local posteriors using MCMC.

We have also observed that FEDAVG is a special case of FEDPA (from the algorithmic point of view), since it can be viewed as also using the Laplace approximation for the posteriors, but estimating local covariances $\hat{\boldsymbol{\Sigma}}_i$'s with identities and local means using the final iterates of local SGD.

In this section, we analyze the effects of approximations 2 and 3 on the convergence of FEDPA. Specifically, we first discuss the convergence rates of FEDAVG and FEDPA as *biased* stochastic gradient optimization methods (Ajalloeian & Stich, 2020). We show how the bias and variance of the client deltas behave for FEDAVG and FEDPA as functions of the number samples. We also analyze the quality of samples produced by IASG (Mandt et al., 2017) and how they depend on the amount of local computation and hyperparameters. Our analyses are conducted empirically on synthetic data.

### A.1  DISCUSSION OF THE CONVERGENCE OF FEDPA VS. FEDAVG

First, observe that if each client is able to perfectly estimate their $\boldsymbol{\Delta}_i = \boldsymbol{\Sigma}_i^{-1}(\boldsymbol{\theta} - \boldsymbol{\mu})$, the problem solved by Algorithm 1 simply becomes an optimization of a quadratic objective using unbiased stochastic gradients, $\boldsymbol{\Delta} := \frac{1}{M}\sum_{i=1}^{M}\boldsymbol{\Delta}_i$. The noise in the gradients in this case comes from the fact that the server interacts with only a small subset of $M$ out of $N$ clients in each round. This is a classical stochastic optimization problem with well-known convergence rates under some assumptions on the norm of the stochastic gradients (*e.g.*, Nemirovski et al., 2009). The rate of convergence for SGD with a $\mathcal{O}(t^{-1})$ decaying learning rate used on the server is $\mathcal{O}(1/\sqrt{t})$. It can be further improved to $\mathcal{O}(1/t)$ using Polyak momentum (Polyak, 1964) or iterate averaging (Polyak & Juditsky, 1992).

In reality, both FEDAVG and FEDPA produce biased estimates $\hat{\boldsymbol{\Delta}}_{\text{FEDAVG}}$ and $\hat{\boldsymbol{\Delta}}_{\text{FEDPA}}$, respectively. Thus, we can analyze the problem as SGD with biased stochastic gradient estimates and let $\hat{\boldsymbol{\Delta}}_t := \nabla F(\boldsymbol{\theta}_t) + \mathbf{b}(\boldsymbol{\theta}_t) + \mathbf{n}(\boldsymbol{\theta}_t)$ where $\mathbf{b}(\boldsymbol{\theta}_t)$ and $\mathbf{n}(\boldsymbol{\theta}_t, \xi)$ are bias and noise terms. Following Ajalloeian & Stich (2020), we can further assume that the bias and noise terms are norm-bounded as follows.

**Assumption 4 ($(m, \zeta^2)$-bounded bias)** *There exist constants $0 \le m < 1$ and $\zeta^2 \ge 0$ such that*

$$\|\mathbf{b}(\boldsymbol{\theta})\|^2 \le m\|\nabla F(\boldsymbol{\theta})\|^2 + \zeta^2, \quad \forall \boldsymbol{\theta} \in \mathbb{R}^d. \tag{5}$$

**Assumption 5 ($(M, \sigma^2)$-bounded noise)** *There exist constants $0 \le M < 1$ and $\sigma^2 \ge 0$ such that*

$$\mathbb{E}_{\xi}\left[\|\mathbf{n}(\boldsymbol{\theta}, \xi)\|^2\right] \le M\|\nabla F(\boldsymbol{\theta})\|^2 + \sigma^2, \quad \forall \boldsymbol{\theta} \in \mathbb{R}^d. \tag{6}$$

Under these general assumptions, the following convergence result holds.

**Theorem 6 (Ajalloeian & Stich (2020), Theorem 2)** *Let $F(\boldsymbol{\theta})$ be $L$-smooth. Then SGD with a learning rate $\alpha := \min\left\{\frac{1}{L}, \frac{1-m}{2ML}, \left(\frac{LF}{\sigma^2 T}\right)^{1/2}\right\}$ and gradients that satisfy Assumptions 4, 5 achieves the vicinity of a stationary point, $\mathbb{E}\left[\|\nabla F(\boldsymbol{\theta})\|^2\right] = \mathcal{O}\left(\varepsilon + \frac{\zeta^2}{1-m}\right)$, in $T$ iterations, where*

$$T = \mathcal{O}\left(\frac{1}{\varepsilon}\left[1 + \frac{M}{1-m} + \frac{\sigma^2}{\varepsilon(1-m)}\right]\right)\frac{LF}{1-m}. \tag{7}$$

Note that SGD with biased gradients is able to converge to a vicinity of the optimum determined by the bias term $\zeta^2/(1 - m)$. For FEDAVG, since the bias is not countered, this term determines the distance between the stationary point and the true global optimum. For FEDPA, since $\hat{\boldsymbol{\Delta}}_{\text{FEDPA}} \to \boldsymbol{\Delta}$ with more local samples, the bias should vanish as we increase the amount of local computation.

Determining the precise statistical dependence of the gradient bias on the local samples is beyond the scope of this work. However, to gain more intuition about the differences in behavior of FEDPA and FEDAVG, below we conduct an empirical analysis of the bias and variance of the estimated client deltas on synthetic least squares problems, for which exact deltas can be computed analytically.

**(a)** FEDAVG bias and variance as functions of the number of local steps.

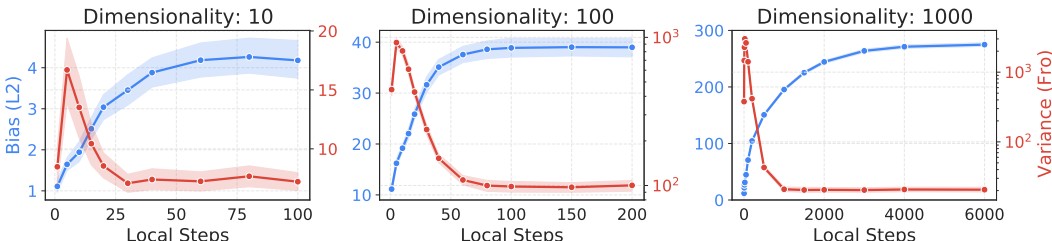

**(b)** FEDPA bias and variance as functions of the number of local steps. The burn-in steps were not included. For dimensionality 10, 100, and 1000, the shrinkage $\rho$ was fixed to 0.01, 0.005, and 0.001, respectively.

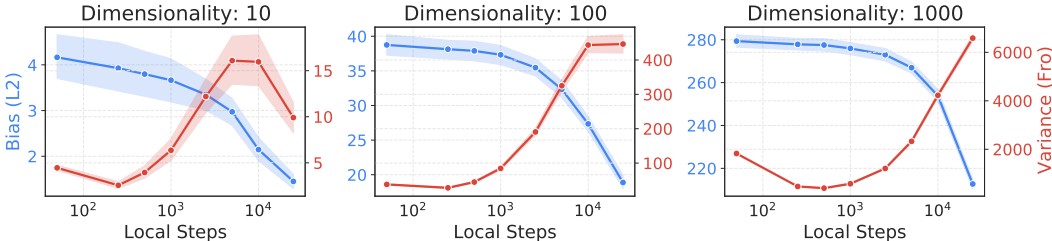

**(c)** FEDPA bias and variance as functions of the shrinkage parameter. For dimensionality 10, 100, and 1000, the number of local steps was fixed to 5,000, 10,000, and 50,000, respectively.

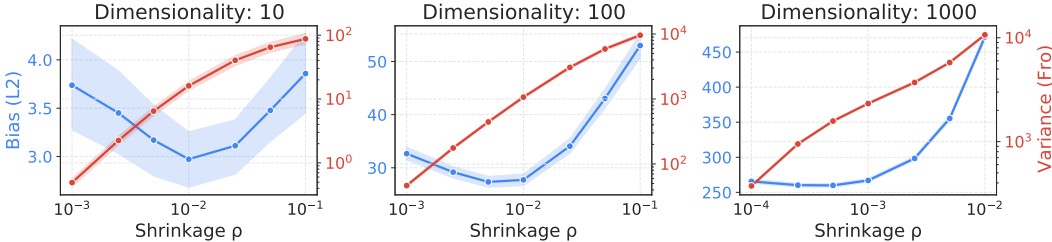

**Figure 3:** The bias and variance tradeoffs for FEDAVG and FEDPA as functions of the estimation parameters.

**Quantifying empirically the bias and variance of $\hat{\Delta}$ for FEDPA and FEDAVG.** We measure the empirical bias and variance of the client deltas computed by each of the methods on the synthetic least squares linear regression problems generated according to Guyon (2003) using the `make_regression` function from scikit-learn.[9] The problems were generated as follows: for each dimensionality (10, 100, and 1000 features), we generated 10 random least squares problems, each of which consisted of 500 synthetic data points. Next, for each of the problems we generated 10 random initial model parameters $\{\boldsymbol{\theta}_1, \ldots, \boldsymbol{\theta}_{10}\}$ and for each of the parameters we computed the exact $\boldsymbol{\Delta}_i$ as well as $\hat{\boldsymbol{\Delta}}_{\text{FEDAVG},i}$ and $\hat{\boldsymbol{\Delta}}_{\text{FEDPA},i}$ for different numbers of local steps; for $\hat{\boldsymbol{\Delta}}_{\text{FEDPA}}$ we also varied the shrinkage hyperparameter. Using these sample estimates, we further computed the $L_2$-norm of the bias and the Frobenius norm of the covariance matrices as functions of the number of local steps.

The results are presented on Fig. 3. From Fig. 3a, we see that as the amount of local computation increases, the bias in FEDAVG delta estimates grows and the variance reduces. For FEDPA (Fig. 3b), the trends turn out to be the opposite: as the number of local steps increases, the bias consistently reduces; the variance initially goes up, but with enough samples joins the downward trend. Note that the initial upward trend in the variance is due to the fact that we used the same *fixed* shrinkage $\rho$ regardless of the number of local steps. To avoid sharp increases in the variance, $\rho$ must be selected for each number of local steps separately; Fig. 3c demonstrates how the bias and variance depend on the shrinkage hyperparameter for some fixed number of local steps.[10]

---

[9] https://scikit-learn.org/stable/modules/generated/sklearn.datasets.make_regression.html

[10] One could also use posterior samples to estimate the best possible $\rho$ that balance the bias-variance tradeoff (*e.g.*, Chen et al., 2010) and avoids sharp increases in the variance.

## A.2 ANALYSIS OF THE QUALITY OF IASG-BASED SAMPLING AND COVARIANCE

The more and better samples we can obtain locally, the lower the bias and variance of the gradients of $\mathcal{Q}(\boldsymbol{\theta})$ will be, resulting in faster convergence to a fixed point closer to the global optimum. For local sampling, we proposed to use a variant of SG-MCMC called Iterate Averaged Stochastic Gradient (IASG) developed by Mandt et al. (2017), given in Algorithm 4. The algorithm generates samples by simply averaging every $K$ intermediate iterates produced by a client optimizer (typically, SGD with some a fixed learning rate $\alpha$) after skipping the first $B$ iterates as a burn-in phase.[11]

*How good are the samples produced by IASG and how do different parameters of the algorithm affect the quality of the samples?* To answer this question, we run IASG on synthetic least squares problems, for which we can compute the actual posterior distribution and measure the quality of the samples by evaluating the effective sample size (ESS, Liu, 1996; Owen, 2013). Given $\ell$ approximate posterior samples $\{\boldsymbol{\theta}_1, \ldots, \boldsymbol{\theta}_\ell\}$, the ESS statistic can be computed as follows:

$$\text{ESS}\left(\{\boldsymbol{\theta}_i\}_{j=1}^\ell\right) := \left(\sum_{j=1}^\ell w_j\right)^2 \bigg/ \sum_{j=1}^\ell w_j^2 \,,$$

where weights $w_j$ must be proportional to the posterior probabilities, or equivalently to the loss.

**Effects of the dimensionality, the number of data points, and IASG parameters on ESS.** The results of our synthetic experiments are presented below in Fig. 4. The takeaways are as follows:

- More burn-in steps (or epochs) generally improve the quality of samples.
- The larger the number of steps per sample the better (less correlated) the samples are.
- The learning rate is the most sensitive and important hyperparameter—if too large, IASG might diverge (happened in the 1000 dimensional case); if too small, the samples become correlated.
- Finally, the quality of the samples deteriorates with the increase in the number of dimensions.

**(a)** ESS as a function of the number of burn-in steps. (Steps per sample: 50.)

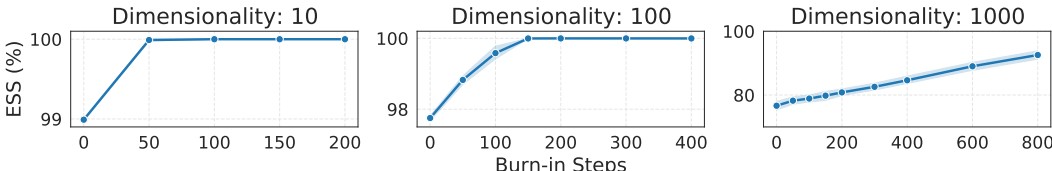

**(b)** ESS as a function of the number of steps per sample. (Burn-in steps: 100.)

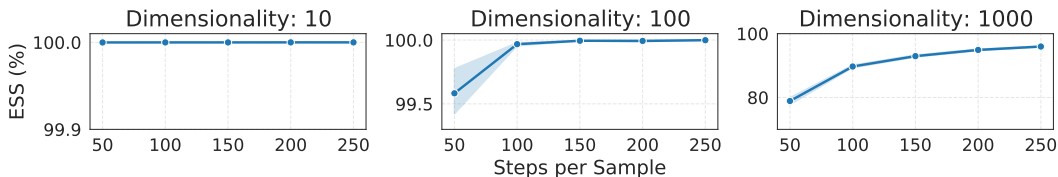

**(c)** ESS as a function of the learning rate. (Burn-in steps: 100, steps per sample: 50.)

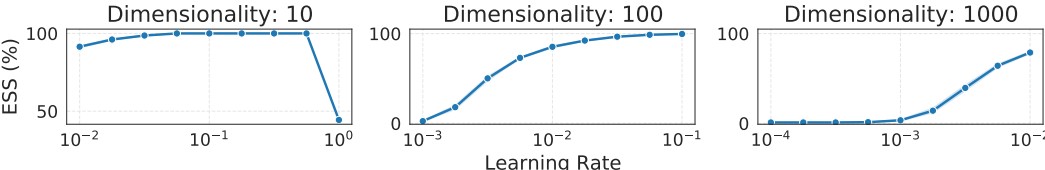

**Figure 4:** The ESS statistics for samples produced by IASG on random synthetic least squares linear regression problems of dimensionality 10, 100, 1000. Total number of data points per problem: 500, batch size: 10. In (a) and (b) the learning rate was set to 0.1 for 10 and 100 dimensions, and 0.01 for 1000 dimensions.

---

[11]Note that in our experiments in Section 5, instead of using $B$ local steps for burn-in at each round, we used several *initial rounds* as burn-in-only rounds, running FEDPA in the FEDAVG regime.

## B   PROOFS

**Proposition 1 (Global Posterior Decomposition)**  *Under the uniform prior, any global posterior distribution that exists decomposes into a product of local posteriors:* $\mathbb{P}\left(\boldsymbol{\theta} \mid D\right) \propto \prod_{i=1}^{N} \mathbb{P}\left(\boldsymbol{\theta} \mid D_i\right).$

**Proof**  Under the uniform prior, the following equivalence holds for $\mathbb{P}\left(\boldsymbol{\theta} \mid D\right)$ as a function of $\boldsymbol{\theta}$:

$$\mathbb{P}\left(\boldsymbol{\theta} \mid D\right) \propto \mathbb{P}\left(D \mid \boldsymbol{\theta}\right) = \prod_{z \in D} \mathbb{P}\left(z \mid \boldsymbol{\theta}\right) = \prod_{i=1}^{N} \underbrace{\prod_{z \in D_i} \mathbb{P}\left(z \mid \boldsymbol{\theta}\right)}_{\text{local likelihood}} \propto \prod_{i=1}^{N} \mathbb{P}\left(\boldsymbol{\theta} \mid D_i\right) \qquad (8)$$

The proportionality constant between the left and right hand side in Eq. 8 is $\prod_{i=1}^{N} \mathbb{P}\left(D_i\right)/\mathbb{P}\left(D\right)$. ∎

**Proposition 2 (Global Posterior Inference)**  *The global posterior mode $\boldsymbol{\mu}$ given in Eq. 3 is the minimizer of a quadratic $\mathcal{Q}(\boldsymbol{\theta}) := \frac{1}{2}\boldsymbol{\theta}^{\top}\mathbf{A}\boldsymbol{\theta} - \mathbf{b}^{\top}\boldsymbol{\theta}$, where $\mathbf{A} := \sum_{i=1}^{N} q_i \boldsymbol{\Sigma}_i^{-1}$ and $\mathbf{b} := \sum_{i=1}^{N} q_i \boldsymbol{\Sigma}_i^{-1} \boldsymbol{\mu}_i$.*

**Proof**  The statement of the proposition (implicitly) assumes that all matrix inverses exist. Then, the quadratic $\mathcal{Q}(\boldsymbol{\theta})$ is positive definite (PD) since $\mathbf{A}$ is PD as a convex combination of PD matrices $\boldsymbol{\Sigma}_i^{-1}$. Thus, the quadratic has a unique solution $\boldsymbol{\theta}^{\star}$ where the gradient of the objective vanishes:

$$\mathbf{A}\boldsymbol{\theta}^{\star} - \mathbf{b} = 0 \quad \Rightarrow \quad \boldsymbol{\theta}^{\star} = \mathbf{A}^{-1}\mathbf{b} = \left(\sum_{i=1}^{N} q_i \boldsymbol{\Sigma}_i^{-1}\right)^{-1} \sum_{i=1}^{N} q_i \boldsymbol{\Sigma}_i^{-1} \boldsymbol{\mu}_i \equiv \boldsymbol{\mu}, \qquad (9)$$

which implies that $\boldsymbol{\mu}$ is the unique minimizer of $\mathcal{Q}(\boldsymbol{\theta})$. ∎

## C   COMPUTATION OF CLIENT DELTAS VIA DYNAMIC PROGRAMMING

In this section, we provide a constructive proof for the following theorem by designing an efficient algorithm for computing $\hat{\boldsymbol{\Delta}}_\ell := \hat{\boldsymbol{\Sigma}}_\ell^{-1}(\boldsymbol{\theta} - \hat{\boldsymbol{\mu}}_\ell)$ on the clients in time and memory linear in the number of dimensions $d$ of the parameter vector $\boldsymbol{\theta} \in \mathbb{R}^d$.

**Theorem 3**  *Given $\ell$ approximate posterior samples $\{\hat{\boldsymbol{\theta}}_1, \ldots, \hat{\boldsymbol{\theta}}_\ell\}$, let $\hat{\boldsymbol{\mu}}_\ell$ be the sample mean, $\hat{\mathbf{S}}_\ell$ be the sample covariance, and $\hat{\boldsymbol{\Sigma}}_\ell := \rho_\ell \mathbf{I} + (1 - \rho_\ell)\hat{\mathbf{S}}_\ell$ be a shrinkage estimator (Ledoit & Wolf, 2004b) of the covariance with $\rho_\ell := 1/(1 + (\ell - 1)\rho)$ for some $\rho \in [0, +\infty)$. Then, for any $\boldsymbol{\theta}$, we can compute $\hat{\boldsymbol{\Delta}}_\ell = \hat{\boldsymbol{\Sigma}}_\ell^{-1}(\boldsymbol{\theta} - \hat{\boldsymbol{\mu}}_\ell)$ in $\mathcal{O}(\ell^2 d)$ time and using $\mathcal{O}(\ell d)$ memory.*

The naïve computation of update vectors (*i.e.*, where we first estimate $\hat{\boldsymbol{\mu}}_\ell$ and $\hat{\boldsymbol{\Sigma}}_\ell$ from posterior samples and use them to compute deltas) requires $\mathcal{O}(d^2)$ storage and $\mathcal{O}(d^3)$ compute on the clients and is both computationally and memory intractable. We derive an algorithm that, given $\ell$ posterior samples, allows us to compute $\hat{\boldsymbol{\Delta}}_\ell$ using only $\mathcal{O}(\ell d)$ memory and $\mathcal{O}(\ell^2 d)$ compute.

The algorithm makes use of the following two components:

1. The shrinkage estimator of the covariance (Ledoit & Wolf, 2004b), which is known to be well-conditioned even in high-dimensional settings (*i.e.*, when the number of samples is smaller than the number of dimensions) and is widely used in econometrics (Ledoit & Wolf, 2004a) and computational biology (Schäfer & Strimmer, 2005).

2. Incremental computation of $\hat{\boldsymbol{\Sigma}}_\ell^{-1}(\boldsymbol{\theta}_\ell - \hat{\boldsymbol{\mu}}_\ell)$ that exploits the fact that each new posterior sample only adds a rank-1 component to $\hat{\boldsymbol{\Sigma}}_\ell$ and applies the Sherman-Morrison formula to derive a dynamic program for updating $\hat{\boldsymbol{\Delta}}_\ell$.

**Notation.**  For the sake of this discussion, we denote $\boldsymbol{\theta}$ (*i.e.*, the server state broadcasted to the clients at round $t$) as $\mathbf{x}_0$, drop the client index $i$, denote posterior samples as $\mathbf{x}_j$, sample mean as $\bar{\mathbf{x}}_\ell := \frac{1}{\ell}\sum_{j=1}^{\ell} \mathbf{x}_j$, and sample covariance as $\hat{\mathbf{S}}_\ell := \frac{1}{\ell-1}\sum_{j=1}^{\ell}(\mathbf{x}_j - \bar{\mathbf{x}}_\ell)(\mathbf{x}_j - \bar{\mathbf{x}}_\ell)^{\top}$.

C.1   THE SHRINKAGE ESTIMATOR OF THE COVARIANCE

Ledoit & Wolf (2004b) proposed to estimate a high-dimensional covariance matrix using a convex combination of identity matrix and sample covariance (known as the LW or shrinkage estimator):

$$\hat{\boldsymbol{\Sigma}}_\ell(\rho_\ell) := \rho_\ell \mathbf{I} + (1 - \rho_\ell)\mathbf{S}_\ell, \tag{10}$$

where $\rho_\ell$ is a scalar parameter that controls the bias-variance tradeoff of the estimator. As an aside, while $\rho_\ell$ can be arbitrary and the optimal $\rho_\ell$ requires knowing the true covariance $\boldsymbol{\Sigma}$, there are near-optimal ways to estimate $\hat{\rho}_\ell$ from the samples (Chen et al., 2010), which we discuss at the end of this section.

In this section, we focus on deriving an expression for $\rho_t$ as a function of $t = 1, \ldots, \ell$ that ensures that the difference between $\hat{\boldsymbol{\Sigma}}_t$ and $\hat{\boldsymbol{\Sigma}}_{t-1}$ is a rank-1 matrix (this is not the case for arbitrary $\rho$'s).

**Derivation of a shrinkage estimator that admits rank-1 updates.**   Consider the following matrix:

$$\tilde{\boldsymbol{\Sigma}}_t := \mathbf{I} + \beta_t \hat{\mathbf{S}}_t, \tag{11}$$

where $\beta_t$ is a scalar function of $t = 1, 2, \ldots, \ell$. We would like to find $\beta_t$ such that $\tilde{\boldsymbol{\Sigma}}_t = \tilde{\boldsymbol{\Sigma}}_{t-1} + \gamma_t \mathbf{U}_t$, where $\mathbf{U}_t$ is a rank-1 matrix, *i.e.*, the following equality should hold:

$$\beta_t \hat{\mathbf{S}}_t = \beta_{t-1} \hat{\mathbf{S}}_{t-1} + \gamma_t \mathbf{U}_t \tag{12}$$

To determine the functional form of $\beta_t$, we need recurrent relationships for $\bar{\mathbf{x}}_t$ and $\hat{\mathbf{S}}_t$. For the former, note that the following relationship holds for two consecutive estimates of the sample mean, $\bar{\mathbf{x}}_{t-1}$ and $\bar{\mathbf{x}}_t$:

$$\bar{\mathbf{x}}_t = \frac{(t-1)\bar{\mathbf{x}}_{t-1} + \mathbf{x}_t}{t} = \bar{\mathbf{x}}_{t-1} + \frac{1}{t}(\mathbf{x}_t - \bar{\mathbf{x}}_{t-1}) \tag{13}$$

This allows us to expand $\hat{\mathbf{S}}_t$ as follows:

$$
\begin{aligned}
(t-1)\hat{\mathbf{S}}_t &= \sum_{j=1}^{t} (\mathbf{x}_j - \bar{\mathbf{x}}_t)(\mathbf{x}_j - \bar{\mathbf{x}}_t)^\top \\
&= \sum_{j=1}^{t} \left(\mathbf{x}_j - \bar{\mathbf{x}}_{t-1} - \frac{\mathbf{x}_t - \bar{\mathbf{x}}_{t-1}}{t}\right)\left(\mathbf{x}_j - \bar{\mathbf{x}}_{t-1} - \frac{\mathbf{x}_t - \bar{\mathbf{x}}_{t-1}}{t}\right)^\top \\
&= \underbrace{\sum_{j=1}^{t-1} (\mathbf{x}_j - \bar{\mathbf{x}}_{t-1})(\mathbf{x}_j - \bar{\mathbf{x}}_{t-1})^\top}_{=(t-2)\hat{\mathbf{S}}_{t-1}} - 2\frac{\mathbf{x}_t - \bar{\mathbf{x}}_{t-1}}{t}\underbrace{\sum_{j=1}^{t-1} (\mathbf{x}_j - \bar{\mathbf{x}}_{t-1})^\top}_{=0} + \\
&\quad \frac{t-1}{t^2}(\mathbf{x}_t - \bar{\mathbf{x}}_{t-1})(\mathbf{x}_t - \bar{\mathbf{x}}_{t-1})^\top + \left(\frac{t-1}{t}\right)^2 (\mathbf{x}_t - \bar{\mathbf{x}}_{t-1})(\mathbf{x}_t - \bar{\mathbf{x}}_{t-1})^\top \\
&= (t-2)\hat{\mathbf{S}}_{t-1} + \frac{t-1}{t}(\mathbf{x}_t - \bar{\mathbf{x}}_{t-1})(\mathbf{x}_t - \bar{\mathbf{x}}_{t-1})^\top
\end{aligned}
\tag{14}
$$

Thus, we have the following recurrent relationship between $\hat{\mathbf{S}}_t$ and $\hat{\mathbf{S}}_{t-1}$:

$$\hat{\mathbf{S}}_t = \left(\frac{t-2}{t-1}\right)\hat{\mathbf{S}}_{t-1} + \frac{1}{t}(\mathbf{x}_t - \bar{\mathbf{x}}_{t-1})(\mathbf{x}_t - \bar{\mathbf{x}}_{t-1})^\top \tag{15}$$

Now, we can plug (15) into (12) and obtain the following equation:

$$\beta_t\left(\frac{t-2}{t-1}\right)\hat{\mathbf{S}}_{t-1} + \frac{\beta_t}{t}(\mathbf{x}_t - \bar{\mathbf{x}}_{t-1})(\mathbf{x}_t - \bar{\mathbf{x}}_{t-1})^\top = \beta_{t-1}\mathbf{S}_{t-1} + \gamma_t\mathbf{U}_t, \tag{16}$$

which implies that $\mathbf{U}_t := (\mathbf{x}_t - \bar{\mathbf{x}}_{t-1})(\mathbf{x}_t - \bar{\mathbf{x}}_{t-1})^\top$, $\gamma_t := \beta_t/t$, and the following telescoping expressions for $\beta_t$:

$$\beta_t = \left(\frac{t-1}{t-2}\right)\beta_{t-1} = \left(\frac{t-1}{t-2} \cdot \frac{t-2}{t-3}\right)\beta_{t-2} = \cdots = (t-1)\beta_2, \tag{17}$$

where we set $\beta_2 \equiv \rho \in [0, +\infty)$ to be a constant. Thus, if we define $\tilde{\boldsymbol{\Sigma}}_t := \mathbf{I} + \rho(t-1)\hat{\mathbf{S}}_t$, then the following recurrent relationships will hold:

$$\tilde{\boldsymbol{\Sigma}}_1 = \mathbf{I},$$

$$\tilde{\boldsymbol{\Sigma}}_2 = \mathbf{I} + \rho\hat{\mathbf{S}}_2 = \tilde{\boldsymbol{\Sigma}}_1 + \frac{\rho}{2}(\mathbf{x}_2 - \bar{\mathbf{x}}_1)(\mathbf{x}_2 - \bar{\mathbf{x}}_1)^\top,$$

$$\tilde{\boldsymbol{\Sigma}}_3 = \mathbf{I} + 2\rho\hat{\mathbf{S}}_3 = \tilde{\boldsymbol{\Sigma}}_2 + \frac{2\rho}{3}(\mathbf{x}_3 - \bar{\mathbf{x}}_2)(\mathbf{x}_3 - \bar{\mathbf{x}}_2)^\top,$$

$$\cdots$$

$$\tilde{\boldsymbol{\Sigma}}_t = \mathbf{I} + (t-1)\rho\hat{\mathbf{S}}_{t-1} = \tilde{\boldsymbol{\Sigma}}_{t-1} + \frac{(t-1)\rho}{t}(\mathbf{x}_t - \bar{\mathbf{x}}_{t-1})(\mathbf{x}_t - \bar{\mathbf{x}}_{t-1})^\top \tag{18}$$

Finally, we can obtain a shrinkage estimator of the covariance from $\tilde{\boldsymbol{\Sigma}}_n$ by normalizing coefficients:

$$\hat{\boldsymbol{\Sigma}}_t := \underbrace{\frac{1}{1 + (t-1)\rho}}_{\rho_t}\mathbf{I} + \underbrace{\frac{(t-1)\rho}{1 + (t-1)\rho}}_{1-\rho_t}\hat{\mathbf{S}}_t = \rho_t\tilde{\boldsymbol{\Sigma}}_t \tag{19}$$

Note that $\hat{\boldsymbol{\Sigma}}_1 \equiv \mathbf{I}$ and $\hat{\boldsymbol{\Sigma}}_t \to \mathbf{S}_t$ as $t \to \infty$.

## C.2 Computing Deltas using Sherman-Morrison and Dynamic Programming

Since $\hat{\boldsymbol{\Sigma}}_\ell$ is proportional to $\tilde{\boldsymbol{\Sigma}}_\ell$ and the latter satisfies recurrent rank-1 updates given in Eq. 18, denoting $\mathbf{u}_\ell := \mathbf{x}_\ell - \bar{\mathbf{x}}_{\ell-1}$, we can express $\hat{\boldsymbol{\Sigma}}_\ell^{-1} = \tilde{\boldsymbol{\Sigma}}_\ell^{-1}/\rho_\ell$ using the Sherman-Morrison formula:

$$\tilde{\boldsymbol{\Sigma}}_\ell^{-1} = \tilde{\boldsymbol{\Sigma}}_{\ell-1}^{-1} - \frac{\gamma_\ell\left(\tilde{\boldsymbol{\Sigma}}_{\ell-1}^{-1}\mathbf{u}_\ell\mathbf{u}_\ell^\top\tilde{\boldsymbol{\Sigma}}_{\ell-1}^{-1}\right)}{1 + \gamma_\ell\left(\mathbf{u}_\ell^\top\tilde{\boldsymbol{\Sigma}}_{\ell-1}^{-1}\mathbf{u}_\ell\right)} \tag{20}$$

Note that we would like to estimate $\hat{\boldsymbol{\Delta}}_\ell := \hat{\boldsymbol{\Sigma}}_\ell^{-1}(\mathbf{x}_0 - \bar{\mathbf{x}}_\ell)$, which can be done without computing or storing any matrices if we know $\tilde{\boldsymbol{\Sigma}}_{\ell-1}^{-1}\mathbf{u}_\ell$ and $\tilde{\boldsymbol{\Sigma}}_{\ell-1}^{-1}(\mathbf{x}_0 - \bar{\mathbf{x}}_\ell)$.

Denoting $\tilde{\boldsymbol{\Delta}}_t := \tilde{\boldsymbol{\Sigma}}_t^{-1}(\mathbf{x}_0 - \bar{\mathbf{x}}_t)$, and knowing that $\mathbf{x}_0 - \bar{\mathbf{x}}_\ell = (\mathbf{x}_0 - \bar{\mathbf{x}}_{\ell-1}) - \mathbf{u}_\ell/\ell$ (which follows from Eq. 13), we can compute $\hat{\boldsymbol{\Delta}}_\ell$ using the following recurrence:

$$\tilde{\boldsymbol{\Delta}}_1 := \mathbf{x}_0 - \bar{\mathbf{x}}_1, \quad \mathbf{v}_{1,2} := \mathbf{x}_2 - \bar{\mathbf{x}}_1, \qquad \text{// initial conditions} \tag{21}$$

$$\mathbf{u}_t := \mathbf{x}_t - \bar{\mathbf{x}}_{t-1}, \quad \mathbf{v}_{t-1,t} := \tilde{\boldsymbol{\Sigma}}_{t-1}^{-1}\mathbf{u}_t \qquad \text{// recurrence for } \mathbf{u}_t \text{ and } \mathbf{v}_{t-1,t} \tag{22}$$

$$\tilde{\boldsymbol{\Delta}}_t = \tilde{\boldsymbol{\Delta}}_{t-1} - \left[1 + \frac{\gamma_t\left(t\mathbf{u}_t^\top\tilde{\boldsymbol{\Delta}}_{t-1} - \mathbf{u}_t^\top\mathbf{v}_{t-1,t}\right)}{1 + \gamma_t\left(\mathbf{u}_t^\top\mathbf{v}_{t-1,t}\right)}\right]\frac{\mathbf{v}_{t-1,t}}{t} \qquad \text{// recurrence for } \tilde{\boldsymbol{\Delta}}_t \tag{23}$$

$$\hat{\boldsymbol{\Delta}}_t = \tilde{\boldsymbol{\Delta}}_t/\rho_t \qquad \text{// final step for } \hat{\boldsymbol{\Delta}}_t \tag{24}$$

Remember that our goal is to avoid storing $d \times d$ matrices throughout the computation. In the above recursive equations, all expressions depend only on vector-vector products except the one for $\mathbf{v}_{t-1,t}$ which needs a matrix-vector product. To express the latter one in the form of vector-vector products, we need another 2-index recurrence on $\mathbf{v}_{i,j} := \tilde{\boldsymbol{\Sigma}}_i^{-1}\mathbf{u}_j$:

$$\mathbf{v}_{1,2} = \mathbf{u}_2, \quad \mathbf{v}_{1,3} = \mathbf{u}_3, \quad \ldots \quad \mathbf{v}_{1,t} = \mathbf{u}_t \qquad \text{// initial conditions} \tag{25}$$

$$\mathbf{v}_{t-1,t} = \left[\tilde{\boldsymbol{\Sigma}}_{t-2}^{-1} - \frac{\gamma_{t-1}\left(\tilde{\boldsymbol{\Sigma}}_{t-2}^{-1}\mathbf{u}_{t-1}\mathbf{u}_{t-1}^\top\tilde{\boldsymbol{\Sigma}}_{t-2}^{-1}\right)}{1 + \gamma_{t-1}\left(\mathbf{u}_{t-1}^\top\tilde{\boldsymbol{\Sigma}}_{t-2}^{-1}\mathbf{u}_{t-1}\right)}\right]\mathbf{u}_t \qquad \text{// Sherman-Morrison} \tag{26}$$

$$= \mathbf{v}_{t-2,t} - \frac{\gamma_{t-1}\left(\mathbf{v}_{t-2,t-1}^\top\mathbf{u}_t\right)}{1 + \gamma_{t-1}\left(\mathbf{v}_{t-2,t-1}^\top\mathbf{u}_{t-1}\right)}\mathbf{v}_{t-2,t-1} \tag{27}$$

$$= \mathbf{v}_{1,t} - \sum_{k=2}^{t-1}\frac{\gamma_k\left(\mathbf{v}_{k-1,k}^\top\mathbf{u}_t\right)}{1 + \gamma_k\left(\mathbf{v}_{k-1,k}^\top\mathbf{u}_k\right)}\mathbf{v}_{k-1,k} \qquad \text{// final expression for } \mathbf{v}_{t-1,t} \tag{28}$$

Now, equipped with these two recurrences, given a stream of samples $\mathbf{x}_1, \mathbf{x}_2, \ldots, \mathbf{x}_t, \ldots$, we compute $\hat{\boldsymbol{\Delta}}_t$ for $t \geq 2$ based on $\mathbf{x}_t$, $\{\mathbf{u}_k\}_{k=1}^{t-1}$, $\{\mathbf{v}_{k-2,k-1}\}_{k=1}^{t-1}$ and $\hat{\boldsymbol{\Delta}}_{t-1}$ using the following two steps:

1. Compute $\mathbf{u}_t$ and $\mathbf{v}_{t-1,t}$ using the second recurrence.

2. Compute $\hat{\boldsymbol{\Delta}}_t$ from $\mathbf{u}_t$, $\mathbf{v}_{t-1,t}$, and $\hat{\boldsymbol{\Delta}}_{t-1}$ using the first recurrence.

For each new sample in the sequence, we repeat the two steps to obtain the updated $\hat{\boldsymbol{\Delta}}_t$ estimate, until we have processed all $\ell$ samples. Note that the first step requires $\mathcal{O}(t)$ vector-vector multiplies, *i.e.*, $\mathcal{O}(td)$ compute, and $\mathcal{O}(d)$ memory, and the second step a $\mathcal{O}(1)$ number of vector-vector multiplies. As a result, the computational complexity of estimating $\hat{\boldsymbol{\Delta}}_\ell$ is $\mathcal{O}(\ell^2 d)$ and the storage needed for the dynamic programming state represented by a tuple $\left( \{\mathbf{u}_k\}_{k=1}^{t-1}, \{\mathbf{v}_{k-2,k-1}\}_{k=1}^{t-1}, \hat{\boldsymbol{\Delta}}_{t-1} \right)$ is $\mathcal{O}(\ell d)$.

**The any-time property of the resulting algorithm.** Interestingly, the above algorithm is online as well as *any-time* in the following sense: as we keep sampling more from the posterior, the estimate of $\hat{\boldsymbol{\Delta}}$ keeps improving, but if stopped at any time, the algorithm still produces the best possible estimate under the given time constraint. If the posterior sampler is stopped during the burn-in phase or after having produced only 1 posterior sample, the returned delta will be identical to FEDAVG. By spending more compute on the clients (and a bit of extra memory), with each additional posterior sample $\mathbf{x}_t$, we have $\hat{\boldsymbol{\Delta}}_t \xrightarrow[t \to \infty]{} \boldsymbol{\Sigma}^{-1}(\mathbf{x}_0 - \boldsymbol{\mu})$.

**Optimal selection of $\rho$.** Note that to be able to run the above described algorithm in an online fashion, we have to select and commit to a $\rho$ before seeing any samples. Alternatively, if the online and any-time properties of the algorithm are unnecessary, we can first obtain $\ell$ posterior samples $\{\mathbf{x}_k\}_{k=1}^{\ell}$, then infer a near-optimal $\hat{\rho}_\star$ from these samples—*e.g.*, using the Rao-Blackwellized version of the LW estimator (RBLW) or the oracle approximating shrinkage (OAS), both proposed and analyzed by Chen et al. (2010)—and then use the inferred $\hat{\rho}_\star$ to compute the corresponding delta using our dynamic programming algorithm.

## D  DETAILS ON THE EXPERIMENTAL SETUP

In this part, we provide additional details on our experimental setup, including a more detailed description of the datasets and tasks, models, methods, and hyperparameters.

### D.1  DATASETS, TASKS, AND MODELS

Statistics of the datasets used in our empirical study can be found in Table 2. All the datasets and tasks considered in our study are a subset of the tasks introduced by Reddi et al. (2020).

**EMNIST-62.** The dataset is comprised of $28 \times 28$ images of handwritten digits and lower and upper case English characters (62 different classes total). The federated version of the dataset was introduced by Caldas et al. (2018), and is partitioned by the author of each character. The heterogeneity of the dataset is coming from the different writing style of each author. We use this dataset for the character recognition task, termed EMNIST CR in Reddi et al. (2020) and the same model architecture, which is a 2-layer convolutional network with $3 \times 3$ kernel, max pooling, and dropout, followed by a 128-unit fully connected layer. The model was adopted from the TensorFlow Federated library: `https://bit.ly/3l41LKv`.

**CIFAR-100.** The federated version of CIFAR-100 was introduced by Reddi et al. (2020). The training set of the dataset is partitioned among 500 clients, 100 data points per client. The partitioning was created using a two-step latent Dirichlet allocation (LDA) over to "coarse" to "fine" labels which created a label distribution resembling a more realistic federated setting. For the model, also following Reddi et al. (2020), we used a modified ResNet-18 with group normalization layer instead of batch normalization, as suggested by Hsieh et al. (2019). The model was adopted from the TensorFlow Federated library: `https://bit.ly/33jMv6g`.

**Table 4:** Selected optimizers for each task. For SGD, $m$ denotes momentum. For Adam, $\beta_1 = 0.9, \beta_2 = 0.99$.

| Hyperparameter | EMNIST-62 | CIFAR-100 | StackOverflow NWP | StackOverflow LR |
|---|---|---|---|---|
| SERVEROPT | SGD ($m = 0.9$) | SGD ($m = 0.9$) | Adam ($\tau = 10^{-3}$) | Adagrad ($\tau = 10^{-5}$) |
| CLIENTOPT | SGD ($m = 0.9$) | SGD ($m = 0.9$) | SGD ($m = 0.0$) | SGD ($m = 0.9$) |
| # clients p/round | 100 | 20 | 10 | 10 |

**Table 5:** Hyperparameter grids for each task.

| Hyperparameter | EMNIST-62 | CIFAR-100 | StackOverflow NWP | StackOverflow LR |
|---|---|---|---|---|
| Server learning rate | \{0.01, 0.05, 0.1, 0.5, 1, 5\} | | \{0.01, 0.05, 0.1, 0.5, 1\} | \{0.1, 0.5, 1, 5, 10\} |
| Client learning rate | \{0.001, 0.005, 0.01, 0.05, 0.1\} | | \{0.01, 0.05, 0.1\} | \{1, 5, 10, 50, 100\} |
| Client epochs | | | \{2, 5, 10, 20\} | |
| FEDPA burn-in | | | \{100, 200, 400, 600, 800\} | |
| FEDPA shrinkage | | | \{0.0001, 0.001, 0.01, 0.1, 1\} | |

**StackOverflow.** The dataset consists of text (questions and answers) asked and answered by the total of 342,477 unique users, collected from https://stackoverflow.com. The federated version of the dataset partitions it into clients by the user. In addition, questions and answers in the dataset have associated metadata, which includes tags. We consider two tasks introduced by Reddi et al. (2020): the next word prediction task (NWP) and the tag prediction task via multi-label logistic regression. The vocabulary of the dataset is restricted to 10,000 most frequently used words for each task (*i.e.*, the NWP task becomes a multi-class classification problem with 10,000 classes). The tags are similarly restricted to 500 most frequent ones (*i.e.*, the LR task becomes a multi-label classification proble with 500 labels).

For tag prediction, we use a simple linear regression model where each question or answer are represented by a normalized bag-of-words vector. The model was adopted from the TensorFlow Federated library: https://bit.ly/2EXjAeY.

For the NWP task, we restrict each client to the first 128 sentences in their dataset, perform padding and truncation to ensure that sentences have 20 words, and then represent each sentence as a sequence of indices corresponding to the 10,000 frequently used words, as well as indices representing padding, out-of-vocabulary (OOV) words, beginning of sentence (BOS), and end of sentence (EOS). We note that accuracy of next word prediction is measured only on the content words and *not* on the OOV, BOS, and EOS symbols. We use an RNN model with 96-dimensional word embeddings (trained from scratch), 670-dimensional LSTM layer, followed by a fully connected output softmax layer. The model was adopted from the TensorFlow Federated library: https://bit.ly/2SoSi3X.

## D.2 METHODS

As mentioned in the main text, we used FEDAVG with adaptive server optimizers with 1 or multiple local epochs per client as our baselines. For each task, we selected the best server optimizer based on the results reported by Reddi et al. (2020), given in Table 4. We emphasize, even though we refer to all our baseline methods as FEDAVG, the names of the methods as given by Reddi et al. (2020) should be FEDAVGM for EMNIST-62 and CIFAR-100, FEDADAM for StackOverflow NWP and FEDADAGRAD for StackOverflow LR. Another difference between our baselines and Reddi et al. (2020) is that we ran SGD *with momentum* on the clients for EMNIST-62, CIFAR-100, and StackOverflow LR, as that improved performance of the methods with multiple epochs per client.

Our FEDPA methods used the same configurations as FEDAVG baselines; moreover, FEDPA and FEDAVG were identical (algorithmically) during the burn-in phase and only different in the client-side computation during the sampling phase of FEDPA.

**Table 6:** The best selected hyperparameters for each task.

| Hyperparameter | EMNIST-62 | CIFAR-100 | StackOverflow NWP | StackOverflow LR |
|---|---|---|---|---|
| Server learning rate | 0.5 | 0.5 | 1.0 | 5.0 |
| Client learning rate | 0.01 | 0.01 | 0.1 | 50.0 |
| Client epochs | 5 | 10 | 5 | 5 |
| FEDPA burn-in | 100 | 400 | 800 | 800 |
| FEDPA shrinkage | 0.1 | 0.01 | 0.01 | 0.01 |

### D.3 HYPERPARAMETERS AND GRIDS

All hyperparameter grids are given in Table 5. The best server and client learning rates were selected based on the FEDAVG performance and used for FEDPA. The best selected hyperparameters are given in Table 6.

## E ADDITIONAL EXPERIMENTAL RESULTS

We provide additional experimental results. As mentioned in the main text, the results presented in Table 3 were selected to highlight the differences between the methods with respect to two metrics of interest: (i) the number of rounds until the desired performance, and (ii) the performance achievable within a fixed number of rounds. A much fuller picture is given by the learning curves of each method. Therefore, we plot evaluation losses, accuracies, and metrics of interest over the course of training. On the plots, individual values at each round are indicated with ×-markers and the 10-round running average with a line of the corresponding color.

**EMNIST-62.** Learning curves for FEDAVG and FEDPA on EMNIST-62 are given in Fig. 5. Fig. 5a shows the best FEDAVG-1E, FEDAVG-5E, and FEDPA-5E models and Fig. 5b shows the best FEDAVG-20E, and FEDPA-20E. Apart from the fact that multi-epoch versions converge significantly faster than the 1-epoch FEDAVG-1E, note that the effect of bias reduction when switching from the burn-in to sampling in FEDPA becomes much more pronounced in the 20-epoch version.

**CIFAR-100 and StackOverflow.** Learning curves for various models on CIFAR-100 and Stack-Overflow tasks are presented in Figs. 6 and 7. The takeaways for CIFAR-100 and StackOverflow NWP are essentially the same as for EMNIST-62—much faster convergence with the increased number of local epochs and visually noticeable improvement in losses and accuracies due to sampling-based bias correction in client deltas after the burn-in phase is over. Interestingly, we see that on StackOverflow LR task FEDAVG-1E clearly dominates multi-epoch methods in terms of the loss and recall at 5, losing in precision and macro-F1. Even more puzzling is the significant drop in the average precision of FEDPA-ME after the switching to sampling, while at the same time a jump in recall and F1 metrics. This indicates that the global model moves to a different fixed point where it over-predicts positive labels (*i.e.*, less precise) but also less likely to miss rare labels (*i.e.*, higher recall on rare labels, and as a result a jump in macro-F1). The reason why this happens, however, is unclear.

**(a)** EMNIST-62: Evaluation loss and accuracy for FEDAVG-1E, FEDAVG-5E, and FEDPA-5E.

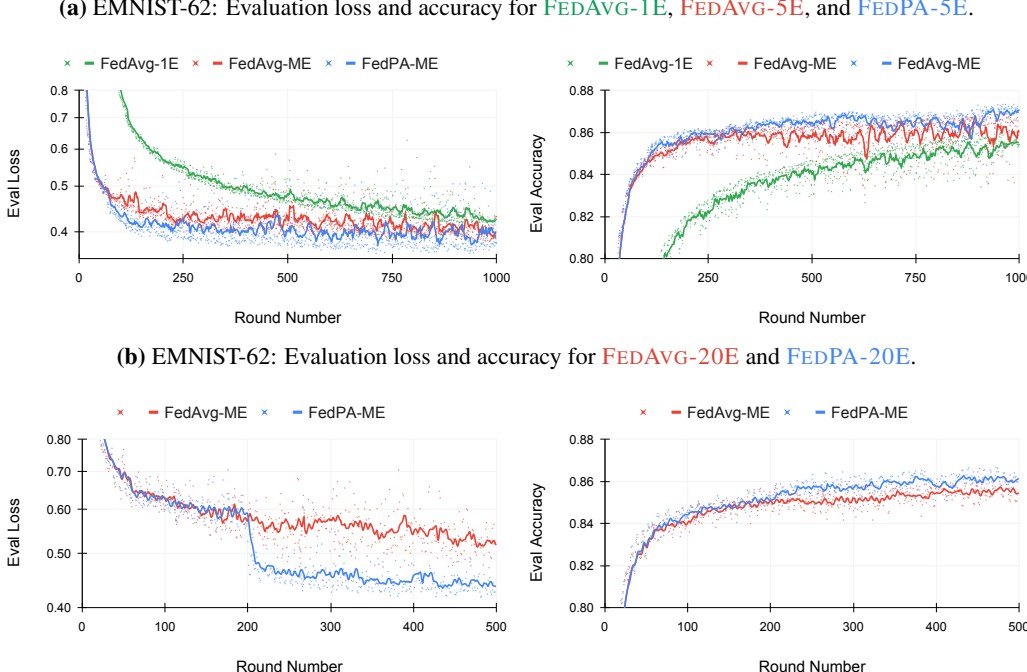

**(b)** EMNIST-62: Evaluation loss and accuracy for FEDAVG-20E and FEDPA-20E.

**Figure 5:** Evaluation metrics for FEDAVG and FEDPA computed at each training round on EMNIST-62.

**(a)** CIFAR-100: Evaluation loss and accuracy for FEDAVG-1E, FEDAVG-10E, and FEDPA-10E.

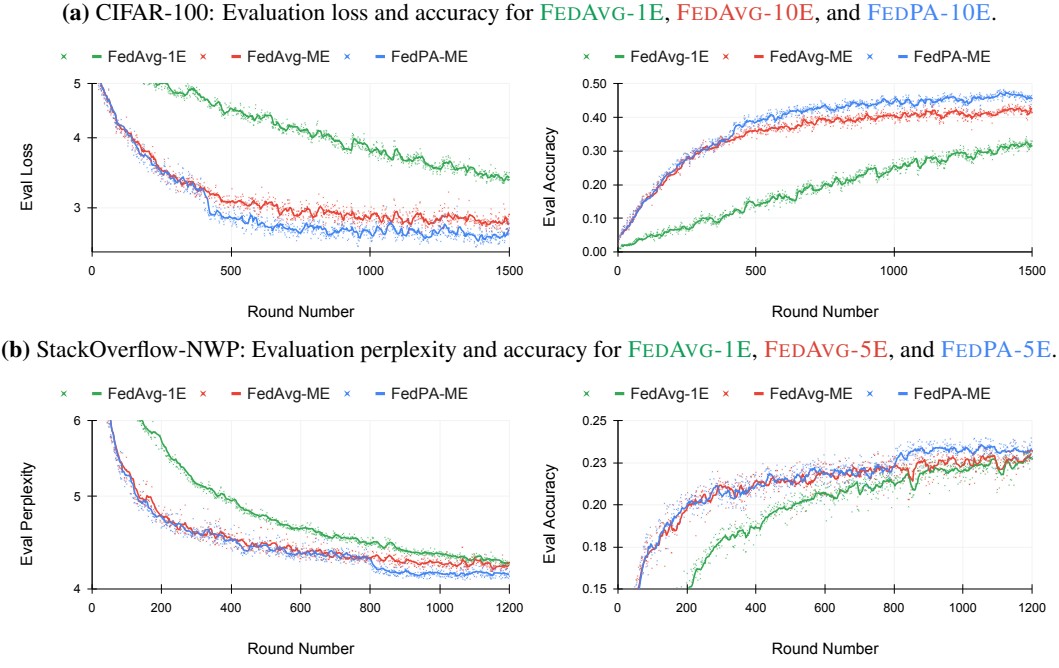

**(b)** StackOverflow-NWP: Evaluation perplexity and accuracy for FEDAVG-1E, FEDAVG-5E, and FEDPA-5E.

**Figure 6:** Evaluation metrics for FEDAVG and FEDPA computed at each training round on (a) CIFAR-100 and (b) StackOverflow NWP tasks.

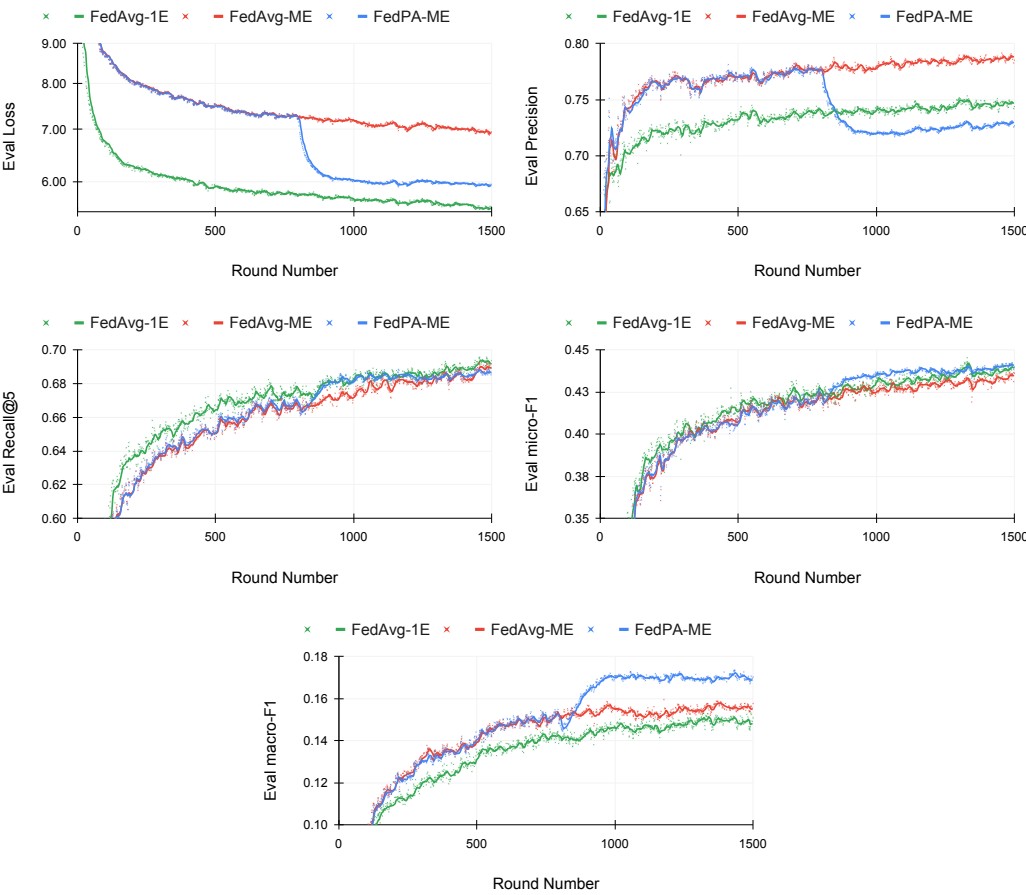

**Figure 7:** Evaluation metrics for FEDAVG and FEDPA computed at each training round on StackOverflow LR. Evaluation loss, average precision and recall, and micro- and macro-averaged F1 for FEDAVG-1E, FEDAVG-5E, and FEDPA-5E.

