# OpenReview forum: "Federated Learning via Posterior Averaging: A New Perspective and Practical Algorithms"
_ICLR.cc/2021/Conference — ICLR 2021 Poster_

### Official Review · AnonReviewer1 · 2020-10-21
**A well motivated and effective method for deriving local updates on the client**

**Rating:** 7
**Confidence:** 3

**Review:**

The authors propose a new method of generating local (client) updates in Federated Learning (FL), where the clients return an adjusted version of their usual local updates to the server. The authors derive this new local update rigorously from the viewpoint of estimating the posterior distribution of the data (under Gaussianity assumptions). They also provide an efficient method for calculating this new update, and show that it outperforms Federated Averaging on several datasets.

Pros:

The new method (FEDPA) is well motivated and is 'as simple as possible, but not simpler' based on the derivation.

FEDPA has standard Federated Averaging (FEDAVG) as a special case, and suggests a family of new methods based on approximations of the covariance matrix, which likely would exhibit bias-variance-tradeoff-esque behaviour.

The authors provide a practical way of calculating the required new quantities efficiently.

Experimental results show FEDPA has superior performance compared to FEDAVG, especially in regimes where client compute is high.

Cons:

The big O analysis of the dynamic programming method for computing the local updates is very useful, but it would also be good to have empirical results on the additional cost on the client. It seems like the cost should not be too significant, but evidence of this would be very valuable, since the cost of FEDPA is strictly greater than the cost of FEDAVG.

The addition of more tuning parameters in FL is never ideal, especially with the knowledge that using too small of a burn in time can lead to arbitrarily bad behaviour. However since the positive effects of FEDPA appear very quickly after the burn in ends, this may be less of a concern in practice.

Would be valuable to include the results of the FEDAVG-1E in the experiments in the main paper, especially since it outperforms both FEDAVG-5E and FEDPA-5E on the StackOverflow LR, which is a surprising and unexpected result.

In all experiments the FEDAVG locally updated using SGD (likely to maintain the connection and comparison with SGD used in IASG). However for FEDAVG, the optimization procedure CLIENTOPT could be something else, such as adam. It would be valuable to know how this FEDPA compares to FEDAVG when the local optimizer is a more powerful method than SGD, since FEDAVG has the freedom to change the local optimizer. (Of course it is only fair to also empirically compare FEDPA where the CLIENTOPT can be the same method. However since the use of SGD in IASG provides it with certain properties, it is less clear if this substitution can be safely made).

The authors provide ample justification for their new method and sufficient evidence that it outperforms the existing standard method FEDAVG.

---

> ### Author Response · Authors · 2020-11-17
> **Thank you for the review and helpful feedback**
>
> Thank you, we appreciate your feedback and the highlights of the positive sides of our paper.
>
> Regarding the Cons:
>
> 1. Thanks for the suggestion to include the empirical analysis of the on-client computation cost. We revised the paper and included an empirical comparison at the end of Section 4 that shows the difference between FedAvg and FedPA in terms of time complexity of the client updates on linear regression tasks of different dimensionality. In practical high-dimensional settings, the difference becomes negligible. We also timed client updates on the benchmark tasks (EMNIST, CIFAR100, StackOverflow), where the difference between FedAvg and FedPA for the same number of local epochs (ranging between 5 and 20) was unnoticeable (i.e., within the margin of error).
>
> 2. Regarding hyperparameter tuning, this is a very good point. Generally, to avoid introducing a new hyperparameter, one could imagine using an “adaptive” algorithm that checks whether a client has “burned-in” (e.g., based on the loss decay) and decides whether to start sampling or to keep running FedPA in the FedAvg regime. Our decision to use a fixed hyperparameter for the # of burn-in steps was motivated by simplicity and the fact that tuning this hyperparameter turned out to be fairly easy in practice on the datasets and tasks we considered.
>
> 3. The results of FedAvg-1E on the very sparse StackOverflow LR task is indeed puzzling. We believe that the effect might be mainly due to extreme sparsity of the problem (the local models were > 95% sparse): more local epochs lead to better local convergence, which made the models more accurate but also hurt the recall of the resulting global model on the held out clients. FedAvg-1E is included in the main text in Table 2.c, and we updated the submission to emphasize this more at the end of Section 5 and urge the reader to check out the full set of figures provided in the appendix.
>
> 4. Regarding local updates for FedAvg and FedPA: this is not quite right, we did actually select different local optimizers (either SGD with or without momentum) depending on which ones worked best for FedAvg and used the same setup for FedPA (see Table 3 in the appendix for details on our setup). Using adaptive optimizers such as Adam for ClientOpt should be possible too, but requires careful treatment of whether and how the local accumulators are treated (e.g., should they be also learned and thus communicated to the server and inferred). Note that Reddi et al. (2020) strictly used SGD for ClientOpt (even without momentum), so we believe that our comparison is fair in that regard. The momentum does not really affect the samples produced by IASG as long as the local learning rates are adjusted, since IASG essentially runs Polyak-averaged SGD which is known to be equivalent to SGD with momentum up to the learning rate (http://proceedings.mlr.press/v40/Flammarion15.pdf).

---

### Official Review · AnonReviewer2 · 2020-10-29
**The paper is nice and seems technically sound. The discussion of the state-of-the-art must be improved.**

**Rating:** 6
**Confidence:** 4

**Review:**

The paper is nice and seems technically sound. However, some part must be clarified.
For instance, the discussion of the state-of-the-art must be improved.
My main concerns is the degree of novelty. It is not clear the difference with other works.
See my comments below.

- Regarding Eq. (1): you state that "For example, least squares loss corresponds to likelihood under a
Gaussian model, cross entropy loss corresponds to likelihood under a categorical model, etc. Thus, Eq. 1 corresponds to maximum likelihood estimation (MLE) of the model parameters ." This sentence must be clarified since it is not straightforward to see it from Eq. (1) and it is an important equation and sentence for your work.  Do you mean that F(\theta) is a log-posterior or a  log-likelihood?
Generally,  it is not a mixture of components. Clarify this point.

- Regarding Eq. (3):  the state-of-the-art related to this equation must be improved. For instance, the following relevant contributions must be considered

Lavancier, F., Rochet, P.: A general procedure to combine estimators. preprint arXiv:1401.6371 (2014)

D. Luengo et al, "Efficient linear fusion of partial estimators", Digital Signal Processing, Volume 78, Pages: 265-283, 2018.

Cattivelli, F.S., Sayed, A.H.: Diffusion LMS strategies for distributed estimation. IEEE Transactions on Signal Processing 58(3), 1035–1048 (2010)

Bordley, R.F.: The combination of forecasts: A Bayesian approach. Journal of the Operational Research Society 33(2), 171–174 (1982)

- It is not clear the difference among your strategy and the consensus one and the others in the papers above. Please clarify.

- Regarding parallel MCMC chains, other schemes could be considered such as:

R. Craiu et al. Learn from thy neighbor: Parallel-chain and regional adaptive MCMC. Journal of the American
Statistical Association, 104(448):1454–1466, 2009.

F. Llorente et al, "Parallel Metropolis-Hastings Coupler", IEEE Signal Processing Letters, Volume 26, Number 6, Pages 953-957, 2019.

J.Corander et al. Parallel interacting MCMC for learning of topologies of graphical models. Data Mining and Knowledge Discovery, 17(3):431–456, 2008

- Regarding the Laplace approximations: you still need the knowledge of the MAP  (maxima a-posteriori). You still need to compute/reach these maxima. Please clarify the computational cost of these previous steps for finding these maxima.

---

> ### Author Response · Authors · 2020-11-17
> **Thank you for the feedback and pointers to the literature**
>
> Thank you for the feedback, we very much appreciate all the pointers to the literature!
>
> Regarding the raised concerns:
>
> 1. **Novelty:** We emphasize that the main focus of this paper is federated learning (FL), not just distributed inference. FL was introduced in 2017 (McMahan et al., 2017), and has not been approached from a distributed inference perspective prior to this work. Our main contributions: (1) showing a connection between FL and distributed inference, (2) designing the first distributed inference algorithm for FL which is practical and tractable from the computation and communication points of view, (3) connect distributed inference with the FedAvg optimization algorithm, which is the default option in FL today. We agree that from the distributed estimation point of view, our method may not be the most advanced possible, but it is the first method designed to work in FL settings (i.e., with limited communication and local computation). Having said that, further improving estimators and sampling techniques is definitely an interesting direction for future work.
>
> 2. **Difference between our method and other consensus-based estimation techniques:**
> Again, our work specifically focuses on FL, while the suggested related work does not directly address the same problem. It would be great to see if other consensus-based algorithms can actually scale to million-dimensional problems and effectively run distributed inference across a very large number of devices. It is not obvious, however, if the suggested consensus-based estimation methods could be adapted to work in federated learning settings.
>
> 3. **Regarding parallel MCMC:** Thanks for your pointers to other MCMC schemes that might hold promise for application to FL. In this work, we did a deep dive into one simple MCMC-based strategy and showed how to make it work within the relatively strict computation and communication efficiency requirements of FL.  We think that it would be an interesting future challenge to see if we could make other MCMC techniques work in a similarly efficient way.
>
> 4. **Regarding the Laplace approximation:** Indeed, we do need to infer local MAP estimates, which is exactly what is accomplished when we use IASG (see Algorithm 4). To see this, note that we use stochastic gradient descent to get close to the optima and, after a number of burn-in steps, we compute a running average of the SGD iterates. This is essentially a Polyak-averaged SGD and it converges to the corresponding optimum (i.e., local MAP estimate) at an accelerated rate.
>
> In a separate comment in this thread, we discuss the suggested references and comment about their relationship to our work.

---

> ### Author Response · Authors · 2020-11-17
> **Summary of the suggested references**
>
> In addition to our response below, we also took a careful look at each of the suggested references and provide 1-sentence summaries for other discussion participants (reviewers, ACs, or anyone interested) with brief comments about the relevance of each of them to our work.
>
> ---
>
> Summary of the suggested references:
> - Lavancier, F., Rochet, P.: A general procedure to combine estimators. preprint arXiv:1401.6371 (2014):
>   - The paper proposes a weighted averaging method for combining multiple estimators of the same quantity (a random vector), where the optimal weights are computed by solving an optimization problem.
>   - It is unclear how this averaging-based estimator is related to Eq. 3 and whether it can be used for posterior inference and/or FL.
> - Bordley, R.F.: The combination of forecasts: A Bayesian approach. Journal of the Operational Research Society 33(2), 171–174 (1982)
>   - Solves a similar problem as the previous reference.
>   - Unclear how it is related to our method and/or FL more broadly.
> - D. Luengo et al, "Efficient linear fusion of partial estimators", Digital Signal Processing, Volume 78, Pages: 265-283, 2018.
>   - The paper derives an optimal way to combine multiple parameter estimators, recovers our Eq. 3 using the minimum mean squared error principle and then provides a couple of alternative estimators.
>   - The connection between FL and the estimation problem in signal processing is nice, but all the proposed estimators seem intractable in very high dimensions.
> - Cattivelli, F.S., Sayed, A.H.: Diffusion LMS strategies for distributed estimation. IEEE Transactions on Signal Processing 58(3), 1035–1048 (2010)
>   - Essentially, provides an iterative algorithm for estimating the global posterior mean (as given in our Eq. 3) but using peer-to-peer communication on a network of interconnected nodes. The approach is focused on linear problems.
>   - FL in peer-to-peer settings should be possible. However, it is beyond the classical server-client communication pattern (hub-and-spoke network topology), which is the focus of our paper.
> - R. Craiu et al. Learn from thy neighbor: Parallel-chain and regional adaptive MCMC. Journal of the American Statistical Association, 104(448):1454–1466, 2009.
> F. Llorente et al, "Parallel Metropolis-Hastings Coupler", IEEE Signal Processing Letters, Volume 26, Number 6, Pages 953-957, 2019.
> J.Corander et al. Parallel interacting MCMC for learning of topologies of graphical models. Data Mining and Knowledge Discovery, 17(3):431–456, 2008
>   - These papers propose different approaches to parallel-chain MCMC with some sort of coupling between the chains.
>   - In our paper, MCMC chains are decoupled and run independently on each client. Coupling MCMC chains would require peer-to-peer communication, which might be interesting and possible in some FL settings, but is not directly relevant to our current work.
>
> ---
>
> For the reference, Eq. 3 from our submission refers to the following: $\mu := \left( \sum_{i=1}^N q_i \Sigma_i^{-1} \right)^{-1} \left( \sum_{i=1}^N q_i \Sigma_i^{-1} \mu_i \right)$.

---

### Official Review · AnonReviewer3 · 2020-10-30
**Review on Federated Learning via Posterior Averaging**

**Rating:** 6
**Confidence:** 2

**Review:**

This paper introduces a new perspective on federated learning through the lens of posterior inference. The paper designs a computation- and communication-efficient posterior inference algorithm—federated posterior averaging (FEDPA), which generalizes FedAvg. FEDPA is compared with the strong baselines in Reddi et al. (2020) on realistic FL benchmarks, which achieves state-of-the-art results with respect to multiple metrics of interest.

Overall, the paper is well written and easy to follow. I tend to accept the paper. The detailed comments follow.

Pros:
1. Viewing federated learning through the lens of posterior inference is new. The motivating example in Figure 1 gives a nice explanation why FedPA may outperform FedAvg.
2. The proposed FedPA algorithm enjoys provable performance guarantee.
3. The simulations are extensive, which demonstrate the clear advantage of FedPA over FedAvg on multiple benchmark tasks.

Cons:
1. The algorithm itself is not well explained in the first eight pages. Consider moving some key equations from appendix to main text to explain clientMCMC.
2. The convergence result is weaker than those on optimization for federated learning. Specifically, no finite-time analysis is provided, and the dependence of the number of local update K on the performance is not clear in the current form.
3. The new challenge relative to distributed MCMC literature is not well explained. Communication? Privacy? or else.

---

> ### Author Response · Authors · 2020-11-17
> **Thank you for the review and feedback**
>
> Thank you for the feedback and for highlighting the positive sides of our paper.
>
> Regarding the Cons:
>
> 1. We agree. Given the extra page provided for the rebuttal and camera ready, we have elaborated the details of the algorithm in the main text (Section 4) and updated the manuscript.
>
> 2. This is correct, our main contributions are primarily algorithmic as well as experimental. Having said that, FedPA generalizes FedAvg, and essentially reduces the bias in the client updates by trading it for some extra variance (coming from local sampling). Assuming that the extra variance is bounded, the existing convergence rates known for FedAvg (e.g., from Reddi et al., 2020) can be directly applicable to FedPA. One could potentially improve these rates by making further assumptions about how exactly bias-variance of the client deltas are affected by our local moment estimation techniques. We have extended Appendix A.1 with a discussion of FedAvg and FedPA convergence as SGD methods with biased gradients and empirical analysis of the bias-variance tradeoff in a synthetic setting (see the updated version of the manuscript). We have also updated the discussion in the main text accordingly.
>
> (Also note that the original paper that introduced FedAvg (McMahan et al., 2017) did not have any formal convergence guarantees; the analysis under a variety of different assumptions was only later developed by the optimization community over the span of a couple of years: https://arxiv.org/abs/1907.02189, https://arxiv.org/pdf/1910.06378.pdf, https://arxiv.org/abs/2002.07839, etc.)
>
> 3. To the best of our knowledge, distributed MCMC techniques---and more broadly inferential approaches---have not been used in the federated learning (FL) context prior to this work. There are two main challenges that we tackle. The first one is indeed communication, which is orders of magnitude more expensive in FL than in the standard distributed optimization and inference settings (FL is designed for low-bandwidth, potentially faulty network, where clients are mobile devices or different organizations). The second is efficient local and global posterior inference in high dimensions (with millions of parameters, as common in deep learning) while keeping the communication cost linear. We will make sure to emphasize this in the paper and discuss the related MCMC literature in more detail. Analysis of the privacy guarantees is left for future work.

---

### Author Response · Authors · 2020-11-17
**Summary of the revision updates**

We would like to thank all reviewers for their feedback and helpful comments.

We have responded to each individual reviewer in the comments below. AnonReviewer2 brought up a few pointers to the literature, and we have additionally provided 1-sentence summaries of all of them in a separate comment for other discussion participants to follow the discussion.

We have also revised and updated the paper and made the following changes:
- Expanded Section 4 with a detailed description of ClientMCMC algorithm (as requested by AnonReviewer3).
- Added an empirical comparison of the computational cost of FedAvg vs. FedPA client updates at the end of Section 4 (as requested by AnonReviewer1).
- Significantly updated Appendix A, which now analyzes FedAvg and FedPA convergence from the perspective of the bias-variance tradeoff in the client updates. The discussion at the end of Section 4 was updated accordingly.

---

### Comment · ~Maruan_Al-Shedivat1 · 2021-03-26
**Supplementary Material**

For everyone who is interested to learn more about FedPA:
- Code (JAX version): https://github.com/alshedivat/fedpa
- Blog post: https://blog.ml.cmu.edu/2021/02/19/an-inferential-perspective-on-federated-learning/
- 60-minute talk: https://www.youtube.com/watch?v=YgDzkqo4Rfo
- 5-minute spotlight: [slideslive link to be added]

---

### Decision · Program_Chairs · 2021-01-07
**Final Decision**

**Decision:**

Accept (Poster)

**Comment:**

The reviewers raised a number of concerns which are addressed by the authors. The paper provides an interesting/novel perspective for federated learning (as a posterior inference problem rather than an optimization problem) which can potentially allow for faster and more accurate solutions.